# VChangeCodec: A High-efficiency Neural Speech Codec with Built-in Voice Changer for Real-time Communication

## Abstract

Neural speech codecs (NSCs) enable high-quality real-time communication (RTC) at low bit rates, making them efficient for bandwidth-constrained environments. However, customizing or modifying the timbre of transmitted voices still relies on separate voice conversion (VC) systems, creating a gap in fully integrated systems that can simultaneously optimize efficient transmission and streaming VC with no additional latency. In this paper, we propose a high-efficiency **VChangeCodec**, which integrates the **V**oice **Change**r model directly into the speech **Codec**. This design seamlessly switches between the original voice mode and customized voice change mode in real-time. Specifically, leveraging the target speaker's embedding, we incorporate a lightweight causal projection network within the encoding module of VChangeCodec to adapt timbre at the token level. These adapted tokens are quantized and transmitted to the decoding module, to generate the converted speech of the target speaker. The integrated framework achieves an ultra-low latency of just 40 ms and requires fewer than 1 million parameters, making it ideal for RTC scenarios such as online conferencing. Our comprehensive evaluations, including subjective listening tests and objective performance assessments, demonstrate that VChangeCodec excels in timbre adaptation capabilities compared to state-of-the-art (SOTA) VC models. We are confident that VChangeCodec provides an efficient and flexible framework for RTC systems, tailored to specific operator requirements.

## 1 Introduction

Speech coding is an essential module in real-time communication (RTC) services. It aims to compress waveforms into representations at a lower bitrate at the sender side and decompress to reconstruct the signal at the receiver side. Recent end-to-end (E2E) neural speech codecs (NSCs) achieve high-quality communication through advanced compression, especially in bandwidth-constrained network environments, enhancing the user experience in various RTC services such as online meetings and voice calls. As mobile live streaming surges in popularity (Chen et al., 2024), users are increasingly interested in modifying their timbre to match personal preferences. Some voice changers (i.e., Conan's bow tie voice changer) have been applied as sound effects processing in live streaming.

The demand for customized voice changers has driven the exploration of voice conversion (VC) technologies, which aims to modify speech style while maintaining linguistic content. Prior works have explored various architectures including Transformers (Tanaka et al., 2019; Kameoka et al., 2020), Auto-encoders (Qian et al., 2019), Generative Adversarial Networks (GAN) (Kaneko et al., 2019; Kaneko & Kameoka, 2018; Nguyen & Cardinaux, 2022),and diffusion models (Popov et al., 2021; Liu et al., 2021a). Nevertheless, their non-streamable architecture and reliance on full utterance inputs severely impede RTC applications.

Later, streaming VC with causal processing is proposed to address these challenges. Recent approaches (Chen et al., 2023; Liu et al., 2021b; Guo et al., 2023; Li et al., 2023; Kovela et al., 2023) adopt pre-trained feature extraction networks (e.g., HuBERT (Hsu et al., 2021) and WavLM (Chen et al., 2022)) to obtain the speech content and use the phoneme-posteriorgram (PPG) (Chen et al., 2023; Liu et al., 2021b; Kovela et al., 2023; Wang et al., 2023) methodology to reconstruct the speech waveform. Additionally, a prevalent strategy (Hayashi et al., 2022; Ning et al., 2024) incorporates

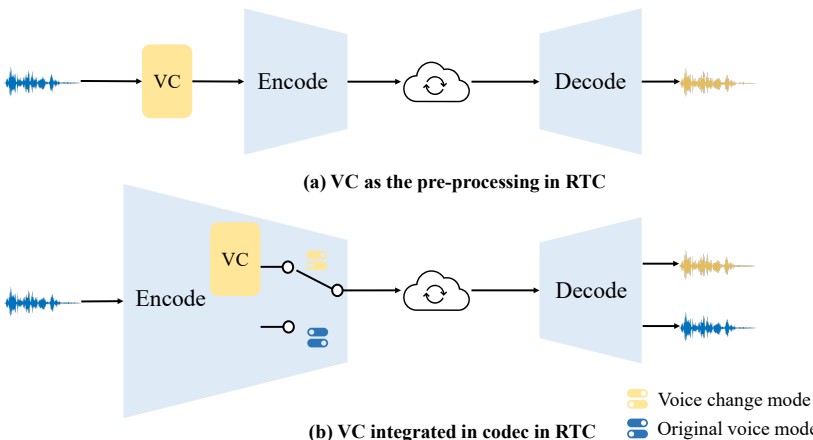

(a) VC as the pre-processing in RTC

(b) VC integrated in codec in RTC

Voice change mode
Original voice mode

Figure 1: Flowchart comparison of the NSC with built-in voice changer solution and existing VC solutions. Blue waveform denotes a source speech and yellow waveform is a target speech. (a) SOTA VC solution as the pre-processing in RTC. (b) Customized voice change only integrated in the encoding module in RTC. **Blue block** represents a universal pre-trained codec. **Yellow block** represents the VC module. The switch determines voice change activation.

teacher models to enhance streaming performance through knowledge transfer. These models offer a potential way to the customized VC. However, traditional VC architectures are typically deployed offline on the user side, by adding a pre-processing system before speech codec. These approaches often face challenges in meeting real-time processing demands due to high latency.

Unlike the existing streaming VC models, we consider the voice changer from the perspective of real-time voice communication. In Figure 1 (a), the mainstream VC model is the pre-processing module in the entire RTC processing chain, requiring a universal codec to transmit the converted waveform. For example, the lightest AC-VC (Ronssin & Cernak, 2021) model exhibits an algorithmic delay of 57.5 ms when running on a CPU infrastructure. Considering the additional 10 ms delay of the LPCNet pitch predictor, and the 40 ms delay of the speech codec, the delay will reach a cumulative latency of 107.5 ms. This significantly exceeds the acceptable latency threshold for RTC requirements. Due to the rapid development of NSCs, it is possible to add the feature of customized voice changer into the codec, directly. An ideal NSC would compress speech into compact tokens rich in all speech information. Specifically, in Figure1 (b), users can switch the voice change mode at any time according to personal preferences at the sender side and generate new adapted timbre tokens. The new token is fed to the decoding networks at the receiver side to reconstruct the altered speech without any changes to the decoding part. This innovative framework marks a paradigm shift from traditional cascaded VC-codec systems to an integrated solution that simultaneously achieves compression and timbre adaptation within a unified pipeline. In addition, we emphasize that our VChangeCodec is tailored for deployment through mobile network operators. In our system, VC is a built-in part of the voice communication module, with internal configurations managed by operators, ensuring users cannot arbitrarily modify settings and thus minimizing privacy risks. Detailed usage scenarios of VChangeCodec in operator networks are provided in Appendix A.1.

Following the pipeline in Figure 1 (b), we propose **VChangeCodec**, a lightweight and low-latency speech codec that integrates the **V**oice **Change**r model into the **Codec** for an operator-oriented network, to mitigate the high complexity and audio artifacts in the recent neural codecs (Pons et al., 2021) such as SoundStream (Zeghidour et al., 2021) and EnCodec (Défossez et al., 2022). Specifically, we use scalar quantization (SQ) to replace residual vector quantization (RVQ) in VChangeCodec. For the voice changer, we propose a lightweight causal projection network in the encoder to perform timbre adaptation on tokens extracted by the pre-trained codec. Then, these adapted tokens are dequantized and decoded to realize a customized voice changer in the decoder. The target timbre customization is achieved using near-parallel training data generated through open-source voice conversion toolkits. We introduce a new commitment loss between the target token and the predicted token. Besides, we keep the generator-discriminator training strategy used in the pre-trained codec.

Consequently, our VChangeCodec can deploy all operations at the encoder side, and support seamless online switching between the original and customized voice changer modes. The comprehensive experiment results indicate that competitive quality is achieved with the lowest delay compared with SOTA models and demonstrate that the tokens from high-quality VChangeCodec preserve intrinsic speech information. **Additionally, we offer speech samples on the demonstration page [1] and encourage readers to listen.** Our contributions are summarized as follows:

- We develop a new lightweight and high-quality speech codec, VChangeCodec, which can integrate customized voice changer directly. Compared to Descript-Audio-Codec (Kumar et al., 2024), the number of parameters is reduced by 70x and achieves comparable quality.

- VChangeCodec can perform the original voice mode and seamlessly switches the customized voice changer by introducing a lightweight causal projection network (Converter). It is noted that, Converter can be easily and efficiently combined with other existing encoder-quantizer-decoder architecture codecs.

- We identify a critical issue in applying VC models in RTC systems due to the high complexity and long latency. Our new framework combines the compression and VC into a single end-to-end model, to fulfill actual technical requirements with latency of 40 ms.

- Our research is operator-oriented, offering the potential to introduce innovative features to existing voice communication systems with minimizing the privacy infringement.

## 2 RELATED WORK

For a discussion and analysis of detailed related work on neural speech compression models and streaming voice conversion (VC), please refer to Appendix A.8.

To our knowledge, neural speech compression has not yet successfully been combined with VC tasks. The earliest work (Strecha et al., 2005) achieves VC directly by re-using the feature of speech codec based on the code excited linear predictive (CELP) (Bessette et al., 2002) to warp the spectral envelope. The StreamVoice in (Wang et al., 2024) employs a low latency streaming codec Audiodec (Wu et al., 2023) as a speaker prompt for the causal context-aware language model. The overall pipeline latency is 124.3 ms on an A100 GPU. StreamVC (Yang et al., 2024b) utilizes SoundStream Zeghidour et al. (2021) and a pre-trained HuBERT Hsu et al. (2021) model to generate pseudo-labels, enabling real-time processing even on mobile devices, and modifies the communication protocol for both transmission and reception by incorporating the target speaker's embedding. However, there is still a lack of integrated solutions that can achieve efficient transmission and real-time VC without additional latency. Our proposed method has the following key differences: 1) We aim to ensure seamless switching between the original voice mode and voice change mode in the RTC services. 2) We design VChangeCodec in such a way that compression and voice changer can be carried out jointly by the same codec. 3) We insert a lightweight causal projection network between the encoder and decoder, allowing us to achieve the conversion of the target speaker's timbre with low latency.

## 3 VCHANGECODEC

The diagram of VChangeCodec is shown in Figure 2. Our VChangeCodec uses the fully causal convolutional encoder-decoder network, that performs temporal downsampling with a pre-defined striding factor. We quantize the latent feature using a scalar quantization (SQ) to reduce complexity in RTC systems. To better understand the workflow, we show the VChangeCodec's network structure and specific training and inference workflows of the original voice mode in Figure 4 in Appendix A.2. For voice changer, we take the metadata of target speaker and the quantized token from SQ as input to a lightweight causal projection network (Converter).

### 3.1 BASIC STRUCTURE OF VCHANGECODEC

**Generator.** The generator is composed of three components, the encoder, the quantization and the decoder. The input signal represented as $x$ with a frame length 20 ms, the signal $x$ is divided into

---

[1]https://anonymous666-speech.github.io/Demo-VChangeCodec/

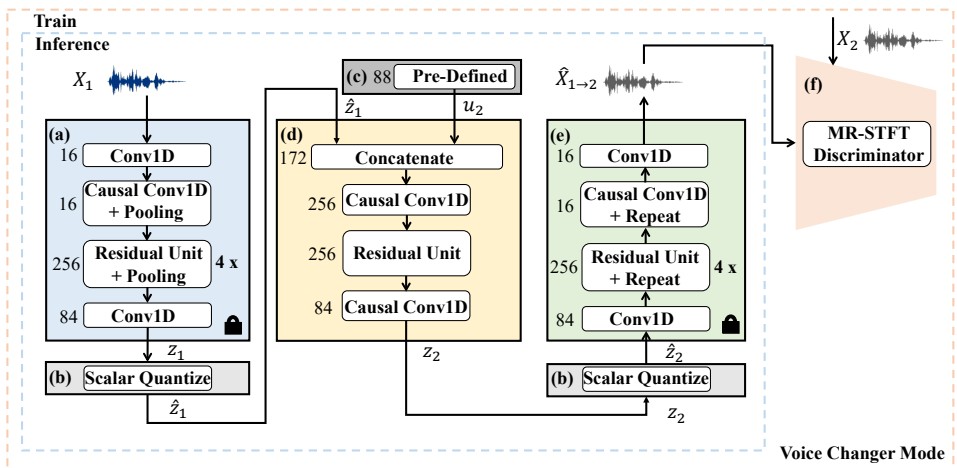

Figure 2: Overview of the proposed VChangeCodec. The number left on each block represents the output dimension of the structure. Causal Conv1D in the encoder and decoder denote the pre-processing layer and post-processing layer. Residual Unit in the encoder and decoder denote the downsampling blocks and upsampling blocks. (a) The encoder of VChangeCodec. (b) The scalar quantization. (c) The pre-defined metadata (deep grey block). (d) The converter network. (e) The decoder of VChangeCodec. (f) The discriminator.

320 bins. The sampling rate is $16000$Hz. The encoder network utilizes a multi-scale downsampling convolutional neural network (CNN) to process the input signal $x$ and distill it into a low-dimensional latent feature $z$. The encoder network consists of a one-dimensional (1D) convolutional layer, a preprocessing layer, and multiple downsampling blocks based on a serial of the dilated CNN and residual unit, and a 1D convolution layer with $\tanh()$ activation function to convert into the latent feature $z \in V_N$, where $V$ is an $N$-dimensional space ($N = 84$). For each frame of 320 samples, the initial 1D convolutional layer extracts intrinsic features from the input signal, yielding an $M$-channel (M=16) feature. The subsequent preprocessing layer, which includes a causal convolution followed by a ReLU activation and average pooling with a downsampling factor of 2, maintains the $M$-channel output. Then, four consecutive downsampling blocks continue the information extraction, and the number of output channels of each downsampling block is 2x of the previous downsampling block. Each downsampling block is composed of four dilated residual units with a dilation rate $d = \{1, 3, 5, 7\}$, and an average pooling by a pre-defined downsampling factor $r_d = \{2, 4, 4, 5\}$. The output feature after four downsampling blocks is $256 \times 1$ with 320x compression.

Residual Vector Quantization (RVQ) in neural codecs can lead to substantial codebooks, thereby increasing the storage demands of RTC services. Inspired by previous work (Mentzer et al., 2023; Yang et al., 2024a), to mitigate high complexity, we introduce a scalar quantization (SQ) to each dimension of $z$ between the encoder network and the decoder network. We claim that our approach is distinct, tailoring the codebook size and implementation to optimize speech codec performance. The SQ discretes the original value with a certain codebook uniformly distributed in $[-1.0, 1.0]$. We set the $R$ to adjust the range of $z$, which can help adapt to the target bitrate. The value of $R$ is 2 in our study. We obtain the value of quantization $\hat{z}$, which is calculated as follows:

$$\hat{z} = \frac{\text{round}(z * R)}{R} \tag{1}$$

For different parameter configurations, the calculation of bit rate can refer to the Appendix A.3.

Regarding to the decoder network, the decoder reconstructs the speech signal using the quantized feature tokens $\hat{z}$. It is a mirror version of the encoder network. The decoder component employs upsampling layers in contrast to the downsampling layers utilized in the encoder. To alleviate calculation complexity, we substitute the transpose convolution with a simpler repeat operation. Additionally, the upsampling rates are applied in the inverse sequence of the downsampling rates. Finally, the final 1D convolution layer is used to generate 320 speech samples.

**Discriminator.** Our adversarial training framework relies on multi-resolution STFT-based (MR-STFT) patch discriminators, which capture spectral structures across varying frequency resolutions. 6 different scales are used with FFT points of $K = \{60, 120, 240, 480, 960, 1920\}$. Each discriminator takes the magnitude spectrum and its logarithmic spectrum is concatenated as input. The discriminators are constructed with seven 2D convolutional layers, with a kernel size of $(3, 3)$.

## 3.2 CUSTOMIZED VOICE CHANGER

**Problem Formulation.** First, a speaker identity $U$ is a random variable drawn from the speaker population $p_U(\cdot)$. Then, an acoustic vector $Z = Z(1 : T)$ is a random process drawn from the joint acoustic distribution $p_Z(\cdot|U)$. In this paper, $Z$ is drawn from the quantized feature token $\hat{z}$. Here acoustic refers to the phonetic, prosodic, content and timbre information and etc. Finally, given the speaker identity and acoustic, the speech segment $X = X(1 : T)$ is a random process randomly sampled from the speech distribution, i.e. $p_X(\cdot|Z(U))$, which characterizes the distribution of the speaker $U$ 's speech uttering the acoustic $Z$. In this paper, we will be working on speech waveform.

Our goal is to design a voice converter that produces the conversion output, $\hat{X}_{1\to2}$, which preserves the acoustic in $X_1$ except timbre information, but matches the speaker characteristics of speaker $U_2$. Formally, an ideal speech converter should have the following desirable property:

$$p_{\hat{X}_{1\to2}}(\cdot|Z(U_2) = \hat{z}_1(u_2)) = p_X(\cdot|Z(U) = \hat{z}_1(u_2)) \tag{2}$$

Eq. (2) means that we replace the speaker information $U_1 = u_1$ in the source speech $Z_1 = z_1$ with the target speaker's identity $U_2 = u_2$, the converted speech should sound like $u_2$ uttering $z_1$.

**Metadata from opensmile.** We follow an open-source implementation [2] to acquire attributes of the target speaker, namely metadata of the target speaker. We extract 88-dimensional acoustic features with openSMILE (Eyben et al., 2010) including $f_0$, loudness, $f_1 - f_3$ frequency, Mel-frequency cepstral coefficient (MFCC), and etc. Specifically, we use the pre-defined feature set of eGeMAPSv02. We anticipate that these acoustic features can represent speaker identity (timbre information) and capture the subtle emotional variations in speech. We do not use pre-trained speaker embeddings such as (Wan et al., 2018), due to concerns about computational costs and additional training overhead. This approach minimizes training burdens, enabling us to concentrate our efforts on VChangeCodec.

**Causal projection network (Converter).** We design a lightweight projection network to achieve timbre adaptation of tokens, which can be conceptualized as a process of "coloring" the source speaker's tokens to resemble those of the target speaker. We utilize the encoder described in section 3.1 to extract discrete tokens, while the decoder is employed for speech generation. The parameters of both encoder and decoder are frozen, meaning we directly load the parameters from the pre-trained codec of the original voice mode. Our subsequent intuitive justification in 5 demonstrates that no further training is necessary to achieve high-quality timbre adaptation of tokens.

As mentioned in Figure 1, we construct the projection network (namely Converter) of voice changer using causal convolutions to enable streaming inference, restricting each output frame to only depend on current and past input frames. Compared to standard convolution, causal convolution shifts padding to precede rather than trail inputs along the time dimension. Specifically, the Converter is composed of three grouped residual units with dilated convolutional layers in Figure 2. We concatenate the metadata of target speaker $u_2$ and quantized tokens $\hat{z}_1$ as input to Converter. So the input channels of the first grouped residual unit is $N + 88$, and the three grouped residual units with dilation rate $d = \{1, 3, 9\}$. The kernel sizes of all convolutional layers are 3, and the number of channels is $128, 256, 128$ successively in the converter. We also use the SQ on the adapted tokens obtained from the three grouped residual units. Finally, the quantized tokens $\hat{z}_2 = \hat{z}_1(u_2)$ are mapped to $N$-channel to input the decoder of VChangeCodec for target speech generation. In particular, our converter network introduces no additional latency and is well compatible with the encoding module and decoding module. Our converter is a plug-and-play module that can be combined with any end-to-end encoder-quantizer-decoder codec.

**Computational latency.** To profile inference latency, we run the encoder, converter and decoder on a single CPU core of a smartphone iPhone X takes 2 ms for each 20 ms chunk of speech. It is tested

---

[2]https://github.com/audeering/opensmile-python

that the entire pipeline can run continuously in real-time in a streaming fashion. The end-to-end latency, a combination of architectural and inference latency, is thus $40+$ ms in this environment.

### 3.3 TRAINING STRATEGY

For training of VChangeCodec, we refer to the generator-discriminator training strategy. We employ a combination of reconstruction loss $\mathbb{L}_{sp}$, adversarial (GAN) loss $\mathbb{L}_{adv}$, feature match loss $\mathbb{L}_{fm}$, perceptual loss $\mathbb{L}_{pe}$. The detailed description is provided in Appendix A.4. For Converter network, we continue to use the multiple loss components mentioned above, with the ground truth being replaced by the target speech. Moreover, we introduce a token commitment loss for timbre adaptation.

**Token commitment loss.** For acquiring the better token adaptation, we design a commitment loss between the ground-truth token of target speech at the encoder, and the predicted token from source speech at the causal projection network. The converter aims to obtain high-quality target speech, so it is desirable for the quantized values obtained from the source speech through the causal projection network and those obtained from the target speech through the encoder to be as close as possible. Token commitment loss is defined as follow:

$$\mathbb{L}_T(x) = \|\hat{z}(x)) - C(\hat{z}(\hat{x}))\|_{L_2} \tag{3}$$

where $\hat{z}(x)$ and $\hat{z}(\hat{x})$ denote the quantized value of source speech at the causal projection network and the quantized value of target speech at the encoder value respectively. $C$ is the Converter network. Therefore, the overall loss is a weighted summation of the above loss functions.

$$\mathbb{L}_{overall}(X) = \lambda_{sp} * \mathbb{L}_{sp} + \lambda_{adv} * \mathbb{L}_{adv} + \\ \lambda_{fm} * \mathbb{L}_{fm} + \lambda_{pe} * \mathbb{L}_{pe} + \lambda_T * \mathbb{L}_T \tag{4}$$

## 4 EXPERIMENTS

### 4.1 SETUP

**Data sources.** For the speech codec VChangeCodec, the training set is divided into two classes. The clean speech is from LibriTTS (Zen et al., 2019), DNS Challenge (Reddy et al., 2020). The mixed speech is generated by combining clean speech and background interference (e.g., noise), including DNS Challenge, MIR-1K (Hsu & Jang, 2009) and FMA (Defferrard et al., 2016). In addition, the training set includes English and Mandarin utterances, all utterances are sampled at 16 kHz. There are 68 independent test utterances including English and Mandarin selected for objective quality measurement. We present additional details of datasets used for training in A.5. For the voice changer, we use VCTK (Veaux et al., 2016) and AISHELL-3 (Shi et al., 2020) as the source utterances, they are internally expressive multi-speaker English and Mandarin corpus. All speech utterances are at a sampling rate of 16 kHz. We select one male and one female speaker from the internal datasets which contain 1-hour data, respectively, to serve as the target timbre. Then we utilize the open source Retrieval-based-Voice-Conversion (RVC) project [3] to construct approximately parallel data. This project yields satisfactory subjective test results for VC. For the test datasets, we select 42 unseen utterances (15 English corpus, 15 Mandarin corpus, 12 internal corpus) from 42 different speakers.

**Metrics of original voice mode.** For evaluating the speech codec, we perform evaluations along four axes: **POLQA, ViSQOL, STOI and DCRMOS**. POLQA (ITU-T, 2011) is selected as the primary objective evaluation metric, which predicts the Mean Opinion Score (MOS) by comparing the spectrum of the reference and degraded signals. The predicted MOS score ranges from $1.0$ to $4.75$, and the average MOS of each system is calculated for evaluation. ViSQOL (Chinen et al., 2020) is an intrusive perceptual quality metric that uses spectral similarity to the ground truth to estimate a MOS. STOI (Taal et al., 2011) shows a high correlation with the intelligibility of noisy and time-frequency weighted noisy speech. Moreover, we organize a subjective listening test by referring to the ITU-T P.800 recommendation (ITU-T, 1996), and the quality evaluation is performed by using the Degradation Category Rating (DCR) method. We select eight Chinese speech utterances and invite 24 native listeners to participate in the listening test.

---

[3]https://github.com/RVC-Project/Retrieval-based-Voice-Conversion-WebUI

**Metrics of voice changer mode.** For evaluating the voice changer, the evaluations are performed along four axes: **naturalness, intelligibility, Mel Cepstral Distortion (MCD)** and **speaker similarity**. Naturalness is rated by DNSMOS (Reddy et al., 2021) which consists of three scores for the quality of speech (SIG), noise (BAK), overall (OVRL), P.808 MOS. Intelligibility is assessed by word and character error rate obtained using the Whisper (Radford et al., 2023) ASR model. Note that the evaluation is conducted only on the English utterances. MCD measures the distance between the Mel-cepstral coefficients of the converted and reference audios. We follow (Guo et al., 2023) by using Resemblyzer [4] to rate speaker similarity. Additionally, we conduct the subjective evaluation of speech naturalness (N-MOS) and speaker similarity (S-MOS).

**Model and training recipe.** We use the AdamW optimizer (Loshchilov & Hutter, 2017) and the Exponential LR scheduler to train the model. For the VChangeCodec of the original mode, the batch size is set to 16. For each training iteration, we randomly select speech clips with a duration of 2 seconds. For the Converter network of voice changer mode, we fix the encoder and decoder of the proposed VChangeCodec. The batch size is set to 8 with two V100 GPUs. The learning rate is set 0.0002. The parameters of the discriminator remain consistent under two modes. The weights $\{\lambda_{sp}, \lambda_{adv} \lambda_{fm}, \lambda_{pe}, \lambda_T\}$ are set to $\{1, 2, 1, 20, 50\}$.

Table 1: Comparison with SOTA speech codec on different metrics. Optimal and suboptimal performance is highlighted. Underline: Optimal performance at low/medium bitrates is underlined.

| METHOD | BITRATE | POLQA ↑ | ViSQOL ↑ | STOI ↑ | # Params (M) |
|---|---|---|---|---|---|
| OPUS (Valin et al., 2012) | 8 kbps | 2.79 | 3.71 | 85.35 | |
| | 10 kbps | 3.46 | 4.15 | 88.99 | – |
| | 16 kbps | 4.29 | 4.46 | 91.96 | |
| EVS (3GPP, 2014) | 7.2 kbps | 3.69 | 3.96 | 95.24 | |
| | 9.6 kbps | 3.89 | 3.87 | 96.28 | – |
| LYRA2 [4] | 6 kbps | 3.45 | 4.12 | 94.82 | $2.4 - 8.4$ |
| | 9.2 kbps | 3.60 | 4.16 | 95.71 | – |
| ENCODEC (Défossez et al., 2022) | 12 kbps | 3.70 | 4.22 | 97.28 | |
| | 24 kbps | 4.06 | 4.23 | 98.02 | – |
| DAC (Kumar et al., 2024) | 8 kbps | 4.30 | 4.43 | 98.25 | 76 |
| VCHANGECODEC (OURS) | 6 kbps ($N = 56$) | 4.02 | 4.40 | 96.81 | 0.88 |
| | 9.5 kbps ($N = 84$) | 4.10 | 4.47 | 97.86 | 0.97 |

## 4.2 QUALITY COMPARISON WITH SOTA SPEECH CODEC

To evaluate the performance of the proposed VChangeCodec, we conduct a comparison with the OPUS (Valin et al., 2012), EVS (3GPP, 2014) and SOTA open-source neural codecs, Lyra2 [4], and Encodec (Défossez et al., 2022) at different bitrates. We also compared it with the most competitive Descript Audio Codec (DAC) (Kumar et al., 2024). For RTC services, a sampling rate of 16kHz is commonly selected since voice is the primary component of the service. We provide a comprehensive description of the these codecs' configuration in the Appendix A.5. The POLQA, ViSQOL and STOI scores of all codecs are illustrated in Table 1. We observe that the proposed VChangeCodec exhibits superior performance compared to OPUS, EVS, Lyra2 and Encodec at similar bitrates. It is noted that the POLQA MOS of VChangeCodec is beyond 4.0, and it even outperforms the Encodec at 24 kbps. Similarly, the ViSQOL score is the highest in all speech codecs. It indicates the merit of our VChangeCodec according to the objective measurement. We compare the parameter size in Table 1 (See complexity analysis in section 4.4). Notably, the comparsion with DAC proves that our VChangeCodec achieves similar performance with much lower parameters (70x reduction).

---

[4]https://github.com/resemble-ai/Resemblyzer
[4]https://opensource.googleblog.com/2022/09/lyra-v2-a-better-faster-and-more-versatile-speech-codec.html

Furthermore, we present the subjective listening test result as illustrated in Figure 3. We observe that the subjective quality of the proposed VChangeCodec is better than other systems under all bitrates. The absolute subjective MOS of VChangeCodec is comparable to OPUS at 16 kbps, which proves the quality of VChange-Codec at low bitrates. In addition, we present subjective evaluation results in Table 9, to compare with other codecs with lower real-time performance (e.g., Encodec, SpeechTokenizer Zhang et al. (2024) and DAC). Our streaming architecture achieves comparable subjective MOS scores to the non-streaming DAC model. Our VChangeCodec can reconstruct audio with high fidelity and free of artifacts and achieve a high

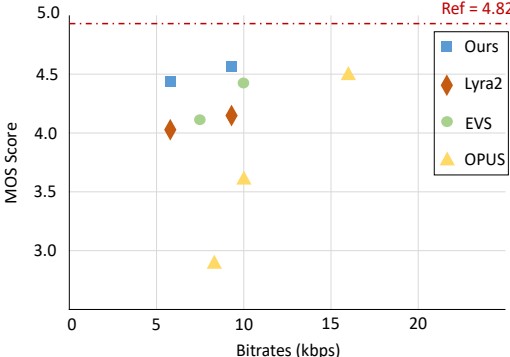

Figure 3: Subjective listening test results.

level of compression to learn a compact token that preserves high-level speech information. This demonstrates VChangeCodec's strong capability in speech reconstruction, which serves as the foundation for voice changer.

Table 2: Comparison with SOTA VC methods. Optimal and suboptimal performance is highlighted. **Our method achieves competitive if not the best performance on all metrics.**

| METHOD | Naturalness ↑ | | | | MCD ↓ | Intelligibility ↓ | | Similarity ↑ |
|---|---|---|---|---|---|---|---|---|
| | SIG | BAK | OVRL | MOS | MCD | WER | CER | Resemblyzer |
| VQMIVC (Wang et al., 2021) | 3.48 | 3.94 | 3.15 | 3.42 | 7.02 | 26.24% | 14.56% | 66.06% |
| Diff-VC (Popov et al., 2021) | 3.31 | 4.12 | 3.08 | 3.70 | 7.81 | 22.60% | 11.15% | 79.74% |
| QuickVC (Guo et al., 2023) | 3.44 | 3.89 | 3.07 | 3.68 | 7.01 | 10.22 % | 4.55 % | 58.33% |
| DDDM-VC Choi et al. (2024) | 2.66 | 3.52 | 2.35 | 3.30 | 7.92 | 42.39% | 25.02% | 78.19% |
| FACodec Ju et al. (2024) | 2.99 | 3.71 | 2.63 | 3.29 | 6.87 | 17.18% | 10.33% | 81.08 % |
| OURS | 3.35 | 4.11 | 3.11 | 3.71 | 5.76 | 16.19 % | 7.67 % | 88.07 % |
| Oracles (Target) | 3.29 | 4.04 | 3.06 | 3.84 | 4.23 | 0.00% | 0.00% | 100.00% |

## 4.3 COMPARISON TO OTHER VC METHODS

**Objective evaluations.** We first select a male timbre for comparative experiments, with the results for the female timbre presented in Appendix A.6. We select five recently proposed VC models capable of one-shot synthesis as the baselines: Diff-VC (Popov et al., 2021), VQMIVC (Wang et al., 2021), QuickVC (Guo et al., 2023), DDDM-VC Choi et al. (2024), FACodec Ju et al. (2024). These solutions are trained on the VCTK, LibriTTS or Librilight dataset, respectively. Baseline VC results are produced by the pre-trained VC models provided in official GitHub repositories. The evaluation results are shown in Table 2. The DNSMOS score shows that VQMIVC, Diff-VC, QuickVC, DDDM-VC, FACodec and our method achieve similar quality, our method acquires the best MOS score and the second-best overall (OVRL) result, outperforming the target speech. Compared with other VC methods, we obtained the lowest MCD score of 5.76, indicating that our spectral reconstruction is close to the target and can successfully convert the spectrogram to the style of ground truth. For intelligibility, QuickVC achieves high performance by incorporating text transcriptions of source utterances, leveraging Hubert-based speech recognition supervision. In contrast, our model operates without text-based supervision, relying solely on acoustic features. Further, our method outperforms other non-streaming VC models. As detailed in Table 7 (in the Appendix A.6), in the case of female timbre, our approach has significantly narrowed this gap. Importantly, we have obtained the best speaker similarity score, surpassing suboptimal FACodec by 6.99%. Our streaming model performs competitive performance on four evaluations especially speaker similarity. These results demonstrate that our approach successfully achieves high-quality timbre adaptation. Our high-efficiency codec which excels in objective performance at low bit rates, is beneficial for the voice changer.

**Subjective evaluations.** To more realistically represent voice quality, we have conducted the subjective evaluation of our voice change mode experiments on speech naturalness and similarity. We evaluated six VC systems, focusing on two target timbres. The test set includes two male and three female speakers. This resulted in five conversion pairs for each specific target timbre, leading to a total of 30 converted utterances from the six VC systems evaluated by each subject. Subjects scored naturalness (N-MOS) and similarity (S-MOS) for 30 converted utterances

Table 3: N-MOS and S-MOS on male timbre

| METHOD | N-MOS | S-MOS | Resemblyzer |
|---|---|---|---|
| VQMIVC (Wang et al., 2021) | 3.24 | 2.18 | 66.06% |
| Diff-VC (Popov et al., 2021) | 2.94 | 2.60 | 79.74% |
| QuickVC (Guo et al., 2023) | **4.05** | 2.60 | 58.33% |
| DDDM-VC Choi et al. (2024) | 2.00 | 2.60 | 78.19% |
| FACodec Ju et al. (2024) | 2.73 | 2.82 | 81.08% |
| OURS | 3.55 | **3.98** | **88.07 %** |

for the specific target timbre. The subjective results are presented in the Table 3. We provide the subjective evaluation of the female timbre in Table 8 (in the Appendix A.6). The results illustrate that our model attains the highest S-MOS scores and the second-best performance in N-MOS. Flow-based QuickVC leads in N-MOS scores and FACodec which relies on disentanglement has a suboptimal performance in S-MOS. Our method is particularly tailored to specific target timbres, offering superior quality in the conversion of selected timbres.

**Retrained SOTA VC models.** To comprehensively evaluate our model's competitiveness, we conducted experiments by retraining selected VC systems with our target timbre dataset. Such experiments can be viewed as an equivalent comparison to any-to-one models. While Diff-VC and FACodec were excluded due to unavailable $f_0$ and spectrogram extraction scripts and training procedures respectively, we focused on three advanced systems: VQMIVC, QuickVC, and the recent DDDM-VC. For retraining details, we trained VQMIVC from scratch for 500 epochs using extracted $f_0$ and spectral features from target male timbre data. We finetuned QuickVC for 3200k steps based on the officially provided model at 1200k steps due to significant training time overhead. We trained DDDM-VC from scratch for 200k steps. The objective evaluation results are presented in Table 4. VQMIVC showed decreased performance, likely due to the need for retraining a dataset-specific vocoder, which needs more complexity. QuickVC demonstrated significant improvement in similarity, though still not reaching our performance. DDDM-VC showed improvements across all objective metrics. As expected, the objective quality of other SOTA methods becomes better. Nevertheless, VChangeCodec maintains superior timbre adaptation capabilities and competitive voice quality.

Overall, VChangeCodec acquires lower latency, better MCD scores, higher speaker similarity and competitive subjective evaluation compared to SOTA methods. These comprehensive improvements make it well-suited for RTC services. Crucially, our VChangeCodec achieves streaming capability and maintains a lightweight architecture of less than one million parameters.

Table 4: Comparison with the retrained/finetuned SOTA VC methods. Optimal and suboptimal performance is highlighted. **Our method achieves competitive performance on all metrics.**

| METHOD | Naturalness ↑ | | | | MCD ↓ | Intelligibility ↓ | | Similarity ↑ |
|---|---|---|---|---|---|---|---|---|
| | SIG | BAK | OVRL | MOS | MCD | WER | CER | Resemblyzer |
| VQMIVC (Wang et al., 2021) | 3.46 | 3.82 | 3.03 | 2.95 | 6.58 | 118.96% | 89.71% | 56.61% |
| QuickVC (Guo et al., 2023) | 3.38 | 4.11 | 3.16 | 3.74 | 6.31 | 9.07 % | 4.96 % | 87.57% |
| DDDM-VC Choi et al. (2024) | 2.21 | 3.28 | 1.96 | 3.34 | 6.73 | 29.49% | 13.97% | 83.00% |
| OURS | 3.35 | 4.11 | 3.11 | 3.71 | 5.76 | 16.19 % | 7.67 % | 88.07% |
| Oracles (Target) | 3.29 | 4.04 | 3.06 | 3.84 | 4.23 | 0.00% | 0.00% | 100.00% |

## 4.4 ABLATION STUDY

We perform a thorough ablation study on our model, systematically varying individual elements of our training strategy and model settings. For model comparison, we employ the four objective metrics detailed in Section 4.1. We conduct ablation studies across four dimensions: metadata, dimensions of the Converter network, loss weights of the token commitment loss, and retraining of the encoder.

For the decoder, we aim for the system to maintain constant parameters at the decoding stage and possess the characteristic of direct decoding. The outcomes of the ablation study are detailed in Table 5. For our ablation study, we train each model with a batch size of 8 and select the model by the best validation performance. Our model uses the metadata from openSMILE as input, the parameter of $\lambda_T$ is 50. The dimension of the Converter is 256 and the encoder is frozen.

Table 5: Results of the ablation study on our proposed VChangeCodec in the voice changer mode.

| | Naturalness ↑ | | | | MCD ↓ | Intelligibility ↓ | | Similarity ↑ |
|---|---|---|---|---|---|---|---|---|
| ABLATION ON | SIG | BAK | OVRL | MOS | MCD | WER | CER | Resemblyzer |
| OURS | **3.35** | 4.11 | **3.11** | 3.71 | **5.76** | **16.19 %** | **7.67 %** | **88.07 %** |
| - wo Metadata | 3.30 | 4.08 | 3.05 | 3.69 | 5.91 | 19.19% | 9.52% | 86.37% |
| - Dims = 128 | 3.18 | 3.94 | 3.02 | 3.62 | 6.12 | 22.38% | 12.78% | 86.76% |
| - Dims = 512 | 3.34 | **4.13** | 3.11 | 3.42 | 5.90 | 20.01% | 11.20% | 88.06% |
| - $\lambda_T = 0$ | 3.24 | 4.07 | 2.98 | 3.68 | 5.93 | 19.30% | 10.25% | 87.76% |
| - $\lambda_T = 30$ | 3.22 | 4.06 | 2.97 | **3.72** | 5.88 | 19.95% | 11.06% | 88.06% |
| - Encoder-tuning | 3.25 | 4.08 | 3.00 | 3.69 | 6.04 | 25.11% | 14.38% | 87.73% |

First, we eliminated metadata to assess its impact and observed a speaker similarity metric of $88.07\%$. Removing metadata led to a roughly $2\%$ decrease in performance, suggesting that indicates that metadata is beneficial to the target timbre because features such as $f_0$ within the metadata play a positive role. Subsequently, we discovered that the Converter dimension significantly influences performance, with lower dimensions consistently yielding inferior metrics. A $128$-dimensional model tends to underperform while increasing the dimensions to $512$ yields a marginal performance improvement. However, considering the number of parameters, we set the dimension to 256. The introduction of a token commitment loss leads to a noticeable enhancement in performance, particularly in terms of target timbre similarity and MCD. Finally, retraining the encoder proves to be unnecessary, as it results in a decline across multiple metrics. The above comprehensive ablation experiments illustrate the optimal parameter configuration of our model.

**Complexity analysis.** We get the number of parameters using PyTorch ptflops, which are listed in Table 1. The parameter sizes of other codec methods come from their original papers. The parameter size of our VChangeCodec is minimal, which is crucial to meet real-time deployment requirements. In contrast, other methods with large parameters may impact their application in the RTC services.

We compare the real-time factor (RTF) over different neural codecs. RTF is defined as the ratio between the temporal length of the input audio and the time needed for the encoder/decoder and converter. We implement our method on a single thread MacBookPro 2021 (Apple M1 Pro chips) and the RTF results are listed in Table 6. The experimental results indicate that the proposed codec outperforms Lyra2 and the increased complexity from the converter is reasonable.

Table 6: Real time factor (RTF) of our method.

| Neural codecs | Encoder | Decoder | Converter |
|---|---|---|---|
| Lyra2 | 0.009 | 0.012 | - |
| **Original voice mode** | **0.007** | **0.007** | - |
| **Voice changer mode** | **0.007** | **0.007** | **0.003** |

## 5 CONCLUSION

We present a novel speech codec framework, VChangeCodec, which seamlessly integrates customized voice changer capabilities directly into its architecture. This integration facilitates real-time switching between the original voice mode and the customized voice change mode. Our approach combines scalar quantization techniques with timbre adaptation using a lightweight causal projection network at the token level. Both subjective and objective evaluations against existing speech codecs demonstrate the superiority of our pre-trained codec model, establishing a promising foundation for voice changers. Extensive experiments validate the advantages of our model over state-of-the-art voice conversion methods, achieving ultra-low latency of $40$ ms for real-time voice conversion. We aim to establish an innovative methodology for voice changers within the real-time communication ecosystem.

## 6 ETHICAL STATEMENT.

Our codec is primarily designed for deployment through an operator's network, rather than for peer-to-peer communication scenarios. Specifically, our lightweight VChangeCodec is embedded within real-time communication systems, where the encoder and decoder are immutable and maintained by the operators. The embedded speaker representation is injected at the sender side and is also maintained by the operators, with the designated timbres being pre-defined and inaccessible to ordinary users. Our method can restrict to a limited target voice, preventing its misuse for impersonating specific targets outside this pre-defined range. We recommend that operators display a notification label on the screen during calls and meetings, such as "Current content is generated by AI!".

## 7 REPRODUCIBILITY STATEMENTS

For the implementation of our model, we provide Figure 2 and a description of the model architecture in Section 3.1 along with the hyper-parameter of the model configuration in Section 4.1. We have shown training and inference processes, and model details in Appendix A.2. To ensure the reproducibility of our experiments, we also share the model details. There are also training loss functions in the Appendix A.4, as well as specific parameter settings. We have uploaded demo samples and we plan to make the inference code public. If our potential legal issues can be resolved, we are prepared to publish the full training implementation for research purposes.

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

# A APPENDIX

## A.1 USAGE SCENARIOS OF VCHANGECODEC

Specifically, our lightweight VChangeCodec is embedded within RTC systems, where the encoder and decoder are immutable and maintained by the operators. This design not only avoids protocol compatibility issues in RTC systems but also minimizes the impact on online services by updating the encoding part only. The embedded speaker representation is injected at the sending end and is also maintained by the operators, with the designated timbres being pre-defined and inaccessible to ordinary users. In contrast, previous voice conversion models, if positioned on the user side, would allow users to arbitrarily modify the pre-defined timbres before they pass through the sender's encoder. The converted timbre, then transmitted through the operator's codec, could raise issues of timbre infringement.

We have carefully considered privacy issues for applications in RTC services. Our VC module is integrated into the speech communication system, prohibiting users from accessing or altering internal configurations. The embedded speaker representation is injected at the sending end and maintained by the operators, with designated timbres being pre-defined and inaccessible to ordinary users. A potential use case is expected the user can only download the related binary file after subscribing to a specific target timbre from the operator. Consequently, the user cannot arbitrarily change the source voice to any target timbre to minimize the deepfake risk to operators.

Zero-shot VC operates on a "one model for all timbres" principle and is not optimized for low latency or low complexity on mobile devices. In contrast, our method adheres to a "one model for one timbre" approach, utilizing a lightweight model tailored for deployment in mid-range performance smartphones. Therefore, our approach is a completely new approach targeting the RTC scenario with distinctive technical requirements and design.

Conclusively, the proposal in this paper aims to enhance the experience in real-time communication, including conversational chatting. Therefore, the extensibility of the target speaker should be properly considered due to privacy concerns. This consideration is a cornerstone of our contribution, and to the best of our knowledge, this represents the first disclosure of a paper addressing this specific aspect.

## A.2 FLOWCHART OF VCHANGECODEC

We show the detailed training and inference workflows for the original voice mode in Figure 4. We also give the internal structure of the Residual Unit. It is composed of three layers of dialted convolution.

## A.3 BITRATE CALCULATION

Given the target bitrate $r$, the dimension of latent feature $N$, the theoretical bitrate in each frame is computed as $-1 * N * \log_2(\frac{1}{2*R+1})$. In this paper, the value of R is 2. For the 84-dimensional codec model, using a codebook size of 5, the bit rate calculation using Shannon's formula is $-1 * 84 * \log_2(\frac{1}{2*2+1}) * 50/1000 = 9.75$ kbps. Since we are considering a uniform distribution where entropy is maximized, the actual bit rate will be lower, at 9.5 kbps.

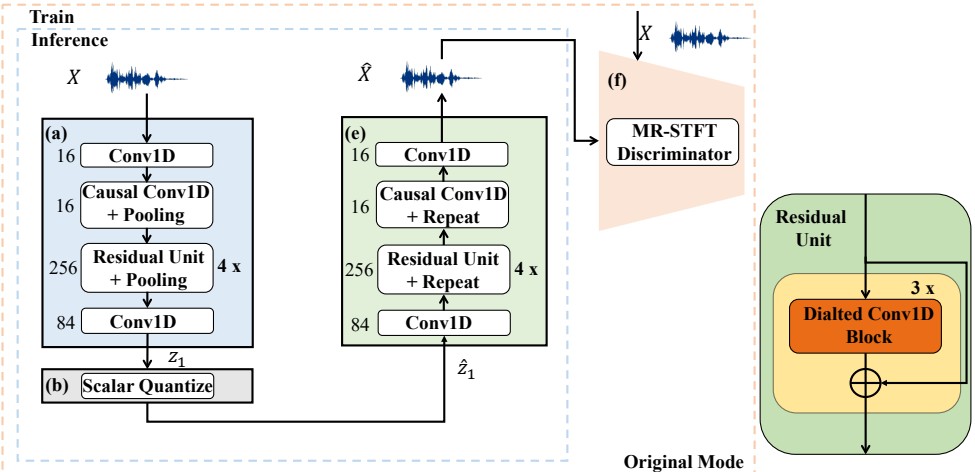

Figure 4: Overview of the proposed VChangeCodec in original mode and the detail of residual unit. (a) The encoder of VChangeCodec. (b) The scalar quantization. (e) The decoder of VChangeCodec. (f) demonstrates the discriminator.

## A.4 TRANING STRATEGY

The training loss contains multiple components.

**Reconstruction loss.** The first one is the reconstruction spectrum loss (Arık et al., 2018). It refers to the reconstruction spectrum loss of multi-resolution STFT (MR-STFT) which targets minimizing the spectrum convergence loss and L1 loss in the logarithmic magnitude spectrum with multiple FFT lengths, which is calculated as follows:

$$\mathbb{L}_{sp}(X) = \sum_d (\| \log(X_d) - log(\hat{X}_d)\|_{L_1} + \frac{\|X_d - \hat{X}_d\|}{\hat{X}_d})$$ (5)

where $X_d$ and $\hat{X}_d$ are the spectrum of ground-truth speech and predicted speech with an FFT length of $2^d$.

**Adversial loss.** Secondly, we also adopt the adversarial training scheme and incorporate it into the training loss function (Mao et al., 2017). The adversarial loss function (generator ($G$)) is defined in:

$$\mathbb{L}_{adv}(x) = \mathbb{E}[(1 - D(G(x)))^2]$$ (6)

where $x$ is the signal in the time domain.

**Feature match loss.** In addition, the feature match loss (Kumar et al., 2019) is appended to minimize the $L_1$ loss between the feature maps of the discriminator for real and generated signal, which is expressed as:

$$\mathbb{L}_{fm}(x) = \mathbb{E}[\frac{1}{L} \sum_{l=0}^{L-1} |D_l(x) - D_l(G(x))|]$$ (7)

where $L$ is the number of layers of the discriminator.

**Perceptual loss.** Then we incorporate the perceptual loss proposed in (Xiao et al., 2023) which evaluates the perceptual loss by comparing the power of the spectrum in equivalent rectangular bandwidth (ERB) of the ground-truth and predicted spectrum, defined in:

$$\mathbb{L}_{pe}(x) = \|P(x) - P(\hat{x})\|_{L_1}$$ (8)

where $P$ is the ERB power of ground-truth spectrum ($x$) and predicted spectrum ($\hat{x}$).

## A.5 EXPERIMENTAL DETAILS

**Datasets details.** Regarding the training set in original voice mode, we use the DNS challenge 2020 dataset and the LibriTTS dataset as the speech part of the training set. Recognizing that neural speech coding is inherently data-driven, we incorporated mixed speech segments, to enhance the robustness of our scheme. Extra mixed speech utterances (mixed with noise or music) are also included, in which the noise clips are from the DNS challenge, and music clips are from MIR-1k and FMA. This configuration is designed for actual RTC scenarios, including pure voice communications, or voice with background interference (e.g. office noise, background music playback, etc). We randomly selected noise crops and adjusted the mixing gain of the noise component using SNR, and the target SNR is 15 dB. $SNR = 10log10(S^2/(kN)^2)$, $S$ denotes clean signal energy and $N$ is noisy signal energy. For all training data, we randomly sample $98\%$ of the dataset for train, $1\%$ for valid and $1\%$ for test. Finally, the unseen test set is strictly an out-of-domain dataset, ensuring it was not exposed to any model during training.

**Details of neural codecs.** For RTC service, 16kHz is the standard sampling rate, as the speech is the primary component in-service. Higher sampling rates are typically for audio (including music), but we focus on speech, operating at a sampling rate of 16kHz. For fair comparison in our experiments, systems (DAC, Encodec, Lyra2) with higher sampling rates were downsampled to 16kHz to ensure consistent evaluation conditions. Specifically, we used the official Encodec versions for 12 kbps ($n_q = 16$) and 24 kbps ($n_q = 32$), with the model's sampling rate at 24 kHz. We downsampled the original test audio from 48kHz to 24 kHz for input into the encoder model and downsampled the output speech to 16 kHz to compare the quality at the same sampling rate. Similarly, for DAC, we used the official configured at 16 kHz, the default 8 kbps model for inference. For Lyra2, we conducted evaluations using the official 16 kHz, default 6 kbps and 9.2 kbps model for inference.

Table 7: Comparison with SOTA VC methods on female timbre. Optimal and suboptimal performance is highlighted. **Our method achieves competitive performance on all metrics.**

| METHOD | Naturalness ↑ | | | | MCD ↓ | Intelligibility ↓ | | Similarity ↑ |
|---|---|---|---|---|---|---|---|---|
| | SIG | BAK | OVRL | MOS | MCD | WER | CER | Resemblyzer |
| VQMIVC (Wang et al., 2021) | 3.44 | 3.87 | 3.07 | 3.15 | 7.52 | 32.43% | 19.85% | 52.70% |
| Diff-VC (Popov et al., 2021) | 3.49 | 4.05 | 3.20 | 3.60 | 8.41 | 30.27% | 14.70% | 70.13% |
| QuickVC (Guo et al., 2023) | 3.53 | 4.11 | 3.27 | 3.58 | 7.37 | 14.32 % | 7.43 % | 47.30% |
| DDDM-VC Choi et al. (2024) | 3.54 | 3.92 | 3.19 | 3.23 | 7.11 | 29.73% | 17.57% | 73.80% |
| FACodec Ju et al. (2024) | 3.50 | 3.99 | 3.23 | 3.40 | 6.78 | 16.20% | 8.29% | 74.11 % |
| OURS | 3.50 | 3.98 | 3.21 | 3.56 | 6.28 | 15.71 % | 8.38 % | 84.80 % |
| Oracles (Target) | 3.54 | 3.94 | 3.20 | 3.70 | 4.89 | 0.00% | 0.00% | 100.00% |

## A.6 COMPARISON TO OTHER VC METHODS ON THE FEMALE TARGET SPEAKER

We conduct the same comparative experiments as described in Section 4.3, with the target female timbre in Table 7. We use the same four metrics for evaluation. The experimental outcomes are largely in alignment with our prior findings, yet it is observable that QuickVC demonstrates superior performance across multiple metrics. However, its speaker similarity performance is comparatively poor. Overall, we have achieved comparable quality to other VC methods. This demonstrates that our voice change mode is capable of delivering personalized voice services.

To make the results more convincing, we conducted subjective evaluations on the target timbre of female in Table 8. The experimental configuration is the same as the male timbre. Based on our experimental results, our model attains the highest scores in subjective evaluation for S-MOS and the near-optimal performance in N-MOS. Consistent with the results for male voices, QuickVC achieves the best naturalness scores, while FACodec ranks second in terms of similarity ratings. While focused on specific target timbres rather than versatile conversion, our method achieves higher quality for the specific timbre.

Table 8: N-MOS and S-MOS on female timbre (Correspondence Table 7)

| METHOD | N-MOS | S-MOS | Resemblyzer |
|---|---|---|---|
| VQMIVC (Wang et al., 2021) | 2.88 | 2.17 | 52.70% |
| Diff-VC (Popov et al., 2021) | 3.02 | 2.05 | 70.13% |
| QuickVC (Guo et al., 2023) | **4.07** | 2.01 | 47.30% |
| DDDM-VC Choi et al. (2024) | 2.71 | 2.36 | 73.80% |
| FACodec Ju et al. (2024) | 3.35 | 2.61 | 74.11% |
| OURS | 4.00 | **4.38** | **84.80 %** |

## A.7  SUBJECTIVE EVALUATIONS ON SOTA NEURAL SPEECH CODECS

Our method addresses Real-Time Communication (RTC) requirements through a fully streaming architecture with low computational complexity. Current implementations of codecs like Descript Audio Codec (DAC) and Encodec face significant limitations in meeting these RTC requirements. Specifically, DAC's architecture, with its 75 million parameters (VChangeCodec only needs less than one million parameters), does not support streaming inference, making it unsuitable for real-time applications. While Encodec does offer streaming capabilities, its processing speed (measured in Real-Time Factor, RTF) is significantly lower than Lyra2, which processes audio approximately 10 times faster in real-time scenarios.

To make our assessment more credible, we have conducted the subjective evaluation results of DAC@8kbps, Encodec@12kbps, and SpeechTokenizer. We selected 10 subjects to conduct DCR-MOS evaluation on four Mandarin corpora. Each subject compared the quality of the four systems and the reference audio, scoring them on a $1-5$ scale. The results are shown in the Table 9. It should be emphasized that we have a streaming structure. Our subjective scores indicate that the competitive quality is achieved with the lowest delay and parameter quantity.

Table 9: Subjective evaluation on different neural speech codecs.

| Neural codecs | VChangeCodec (Ours) | DAC | Encodec | SpeechTokenizer |
|---|---|---|---|---|
| **Bitrate** | 9.5kbps | 8kbps | 12kbps | - |
| **MOS** | 4.54 | **4.55** | 3.52 | 3.74 |

## A.8  RELATED WORK

**Neural speech compression models.**  The VQ-VAEs (Van Den Oord et al., 2017) is a dominant paradigm to train NSCs (Gârbacea et al., 2019), which adopts a convolutional encoder and an autoregressive wavenet (Van Den Oord et al., 2016) decoder. SoundStream (Zeghidour et al., 2021) incorporates the encoder-decoder network and residual vector quantizer (RVQ), combining adversarial and reconstruction losses to achieve excellent generation quality and supporting streamable inference on a smartphone CPU. Encodec (Défossez et al., 2022) uses a multiscale STFT-based (MS-STFT) discriminator to reduce artifacts and produced high-quality samples. They introduce a loss balancer to stabilize training based on the varying scale of gradients coming from the discriminator. Descript-audio-codec (Kumar et al., 2024) can achieve 90x compression with minimal loss in quality and fewer artifacts by improved RVQGAN. However, existing neural speech coding Zhang et al. (2024); Du et al. (2023) models rely on higher parameter quantities to train neural networks to ensure speech quality. Our VChangeCodec adopts scalar quantization instead of RVQ, which enables lighter streaming inference and maintains high fidelity at lower bitrates.

**Streaming voice conversion (VC).**  Diff-VC (Popov et al., 2021) presents a scalable high-quality method based on diffusion probabilistic modeling and considers real-time applications by developing a faster forward Stochastic Differential Equations solver. VQMIVC (Wang et al., 2021) employs vector quantization (VQ) for content encoding and introduces mutual information (MI) as correlation

metrics to achieve disentanglement of content, speaker and pitch representations. QuickVC (Guo et al., 2023) proposes a lightweight VC model based on faster VITS (Kim et al., 2021) and uses HuBERT-soft model to extract content information features. Recent solution (Chen et al., 2023) tackles the streaming VC problem, but the inference latency of the entire streaming VC pipeline is 270 ms on a desktop CPU. However, all these solutions require long latency, making it difficult to apply them in RTC scenarios. Our voice changer based on VChangeCodec can be implemented on a smartphone with a low inference latency of $40+$ ms.

