# OpenReview forum: "VChangeCodec: A High-efficiency Neural Speech Codec with Built-in Voice Changer for Real-time Communication"
_ICLR.cc/2025/Conference — Submitted to ICLR 2025_

### Official Review · Reviewer_2ct9 · 2024-10-28

**Soundness:** 2
**Presentation:** 3
**Contribution:** 2
**Rating:** 5
**Confidence:** 4

**Summary:**

The paper presents VChangeCodec, a speech codec that integrates voice changing capabilities directly into the codec architecture, aimed at enhancing RTC services.

The authors argue that existing neural speech codecs do not support customizable voice features effectively, particularly in bandwidth-constrained environments.

**Strengths:**

1. The reported inference latency of around 40+ ms is impressive and suitable for real-time applications, which is a critical requirement in RTC systems.
2. This paper is well-organized and easy to read.

**Weaknesses:**

1. In Section 4.1, the author claims that ''We select one male and one female speaker from the internal datasets which contain 1-hour data, respectively, to serve as the target timbre.'', I believe the evaluation is not comprehensive and it would be nice to add more target timbre.
2. Some codec baseline systems need to be replaced, the author claims comparison with SOTA codec models in Table 1, while some baselines are proposed in 2012 or 2014, it is not convincing.
3. The author should present the difference between VChangeCodec and two related works [1,2], they also are codec models and can achieve voice conversion. A comparison in experimental evaluation is necessary.
4. A subjective evaluation of the proposed system would be very beneficial. It has been shown time and time again that the opinion of human listeners cannot be replaced with objective evaluation.
5. VC baselines in Table 2 are not SOTA models, please revise the claim or compare the proposed system with the recent SOTA models like LM-VC, SEFVC, or DDDM-VC.

[1] SpeechTokenizer: Unified Speech Tokenizer for Speech Large Language Models

[2] NaturalSpeech 3: Zero-Shot Speech Synthesis with Factorized Codec and Diffusion Models

**Questions:**

1. What advantages does VChangeCodec offer over existing state-of-the-art neural speech codecs in terms of parameter efficiency and compression quality?
2. How can VChangeCodec ensure robust performance across various network conditions typical in RTC scenarios?

---

> ### Author Response · Authors · 2024-11-19
> **Response to Reviewer 2ct9 (1/4)**
>
> Dear Reviewer 2ct9,
>
> We appreciate your positive feedback and find the questions and suggestions to be extremely valuable and constructive! Let us respond to your questions point by point.
>
> > Q1: _**I believe the evaluation is not comprehensive and it would be nice to add more target timbre.**_
>
> First, we acknowledge the reviewer's concerns, which are indeed valuable. The key difference between our approach and traditional voice conversion (VC) models lies in their target audiences and operational contexts. Our lightweight VChangeCodec is embedded within real-time communication systems, where the encoder and decoder are immutable and maintained by mobile network operators.
>
> Our method adheres to a "one model for one timbre" approach, utilizing a lightweight model tailored for deployment in mid-range performance smartphones. Therefore, our approach is a completely new approach **targeting the RTC scenario with distinctive technical requirements and design**.
>
> Secondly, we have carefully considered **privacy issues for applications in RTC services**.  Our VC module is integrated into the speech communication system, prohibiting users from accessing or altering internal configurations.
> The embedded speaker representation is injected at the sending end and is also maintained by the operators, with designated timbres being pre-defined and inaccessible to ordinary users. A potential use case is expected the user can only download the related binary file after subscribing to a specific target timbre from the operator. Consequently, the user cannot change the source voice to any target timbre arbitrarily to minimize the deepfake risk to operators.
> In contrast, previous VC models, if positioned on the user side, would allow users to arbitrarily modify the pre-defined timbres before they pass through the sender's encoder. The converted timbre, then transmitted through the operator's codec, could raise issues of timbre infringement.
>
> Importantly, we have provided additional clarification of "Regarding the differences with zero-shot VC solutions and the limits of target timbre" in **general response**. We hope that the reviewers will take time to peruse these insights. Additionally, we have added **a new target timbre 2 from the open-source VCTK dataset in a demonstration page**. **We would greatly appreciate your time in comparing the subjective quality at the following link**:
>
> https://anonymous666-speech.github.io/Demo-VChangeCodec/
>
> > Q2: _**Some codec baseline systems need to be replaced, the author claims comparison with SOTA codec models in Table 1, while some baselines are proposed in 2012 or 2014, it is not convincing.**_
>
> We appreciate your observation regarding the selection of baseline systems. We understand your concern that some of the baseline systems mentioned in Table 1 including Opus 2012 and EVS 2014. In this section, we supplement additional material to explain why we incorporated OPUS and EVS into the comparison. We also have appended an extra comparison with the SOTA neural codec (DESCRIPT AUDIO CODEC, DAC, Neurisp 2024).
>
> - First, the **RTC speech coding standards widely applied in telecommunication and internet society are Opus (2012) and EVS (2014)** (traditional signal processing-based) with high quality of experience including high quality, low latency and low complexity. And current voice calls and video conferencing services have been widely utilizing these two standards.
> To substantiate that our proposed method is competitive in absolute high quality with higher compressional efficiency, we have compared it with these established standards. A gentle reminder that other neural coding methods, such as Soundstream [1] and Encodec [2], have similarly been benchmarked against these established standards in their experiments.
>
> - Secondly, the DAC currently represents the state-of-the-art (SOTA) performance in the neural speech coding field. Our comprehensive objective evaluation in Table 1 provides a rigorous comparison.
> We utilized POLQA/P.863 metrics, recommended by the International Telecommunication Union (ITU) and recognized as more comprehensive than traditional PESQ. Please refer to the official statement by ITU-T (https://www.itu.int/rec/t-rec-p.862).
> According to the empirical result, we have achieved comparable quality while reducing model parameters by **a remarkable 70x** in Table 4.
> These objective evaluations have validated the effectiveness of our lightweight, streamable codec design for real-time communication scenarios.
>
> Furthermore, we have incorporated the **two systems including NaturalSpeech 3 (FACodec, ICML 2024) and SpeechTokenizer (ICLR 2024)  as suggested in Q3 and Q4**.
>
> [1] Zeghidour N, Luebs A, Omran A, et al. Soundstream: An end-to-end neural audio codec[J]. IEEE/ACM Transactions on Audio, Speech, and Language Processing, 2021, 30: 495-507.
>
> [2] Défossez A, Copet J, Synnaeve G, et al. High fidelity neural audio compression[J]. arXiv preprint arXiv:2210.13438, 2022.

---

> ### Author Response · Authors · 2024-11-19
> **Response to Reviewer 2ct9 (2/4)**
>
> > Q3: _**The author should present the difference between VChangeCodec and two related works inlcuding SpeechTokenizer and NaturalSpeech 3, they also are codec models and can achieve voice conversion. A comparison in experimental evaluation is necessary**_.
>
> First, we must emphasize our innovations in neural codec field and operator-oriented backgrounds.
> Our VChangeCodec is **a lightweight, low-latency codec tailored for mobile network operators**.
> We target deploying the codec through an operator rather than via peer-to-peer communication (as detailed in lines 96-101).
> Our method has strong RTC features (employing a fully streaming architecture) to fulfill the requirements in low complexity, which is not achievable with codecs like Descript Audio Codec (DAC, Neurips 2024), NaturalSpeech 3 (FACodec, ICML 2024), and SpeechTokenizer (ICLR 2024) in their current designs.
> For example, the DAC codec has 75 million parameters, and the FACodec has 1 billion parameters.
>
> Moreover, these highly complex codecs lack a streaming structure, rendering them unsuitable for real-time communication scenarios. Their computational make them impractical for deployment on resource-constrained devices like smartphones. Our design specifically addresses these constraints by providing a lightweight, streaming-compatible solution optimized for mobile platforms.
>
> To enhance the credibility of our results, we conducted **objective evaluations by comparing our method with the NaturalSpeech 3 (FACodec) system for two specific target timbres**. This comparative analysis provides a comprehensive assessment of VC performance across different systems. The objective evaluation results are presented in the table below. Compared with the latest DDDM-VC and FACodec, our objective quality still remains competitive. Bold indicates the best result, and italics indicates the second best result.
>
> Table Comparison with SOTA VC methods on Male timbre (Correspondence Table 2)
>
> |                     |   SIG    |   BAK    |   OVRL   |   MOS    |   MCD    |    WER     |    CER    | Resemblyzer |
> | :-----------------: | :------: | :------: | :------: | :------: | :------: | :--------: | :-------: | :---------: |
> |       VQMIVC        | **3.48** |   3.94   | **3.15** |   3.42   |   7.02   |   26.24%   |  14.56%   |   66.06%    |
> |       Diff-VC       |   3.31   | **4.12** |   3.08   |  _3.70_  |   7.81   |   22.60%   |  11.15%   |   79.74%    |
> |       QuickVC       |  _3.44_  |   3.89   |   3.07   |   3.68   |   7.01   | **10.22%** | **4.55%** |   58.33%    |
> |       DDDM-VC       |   2.66   |   3.52   |   2.35   |   3.30   |   7.92   |   42.29%   |  25.02%   |   78.19%    |
> |       FACodec       |   2.99   |   3.71   |   2.63   |   3.29   |  _6.87_  |   17.18%   |  10.33%   |  _81.08%_   |
> | VChangeCodec (Ours) |   3.35   |  _4.11_  |  _3.11_  | **3.71** | **5.76** |  _16.19%_  |  _7.67%_  | **88.07%**  |
> |  Oracles (Target)   |   3.29   |   4.04   |   3.06   |   3.84   |     -     |      -      |     -      |    100%     |
>
>
> Table Comparison with SOTA VC methods on Female timbre (Correspondence Table 6)
>
> |                     |   SIG    |   BAK    |   OVRL   |   MOS    |   MCD    |    WER     |    CER    | Resemblyzer |
> | :-----------------: | :------: | :------: | :------: | :------: | :------: | :--------: | :-------: | :---------: |
> |       VQMIVC        |   3.44   |   3.87   |   3.07   |   3.15   |   7.52   |   32.43%   |  19.85%   |   52.70%    |
> |       Diff-VC       |   3.49   |  _4.05_  |   3.20   | **3.60** |   8.41   |   30.27%   |  14.70%   |   70.13%    |
> |       QuickVC       |  _3.53_  | **4.11** | **3.27** |  _3.58_  |   7.37   | **14.32%** | **7.43%** |   47.30%    |
> |       DDDM-VC       | **3.54** |   3.92   |   3.19   |   3.23   |   7.11   |   29.73%   |  17.57%   |   73.80%    |
> |       FACodec       |   3.50   |   3.99   |   3.23   |   3.40   |  _6.78_  |   16.20%   |  _8.29%_  |  _74.11%_   |
> | VChangeCodec (Ours) |   3.50   |   3.98   |  _3.21_  |   3.56   | **6.28** |  _15.71%_  |   8.38%   | **84.80%**  |
> |  Oracles (Target)   |   3.54   |   3.94   |   3.20   |   3.70   |     -     |     -       |     -      |    100%     |

---

> > ### Author Response · Authors · 2024-11-19
> > **Response to Reviewer 2ct9 (3/4)**
> >
> > > Q4: _**A subjective evaluation of the proposed system would be very beneficial. It has been shown time and time again that the opinion of human listeners cannot be replaced with objective evaluation**_.
> >
> > Regarding the subjective evaluation of Codec, we have supplemented the subjective evaluation results of DAC@8kbps, Encodec@12kbps, and SpeechTokenizer. We invited ten subjects to conduct a DCRMOS evaluation on four Mandarin corpora. Each subject compared the quality of the four systems' audio and the reference audio, scoring them on a 1-5 scale.
> >
> > The results are shown in the following table. It should be emphasized that we have a streaming structure. Our subjective scores indicate that the competitive quality is achieved with the lowest delay and parameter quantity.
> >
> > Table Subjective evaluation on different neural speech codecs
> > | **Codecs**  | VChangeCodec (Ours) |  DAC  | Encodec | SpeechTokenizer |
> > | :---------: | :----------: | :---: | :-----: | :-------------: |
> > | **Bitrate** |   9.5kbps    | 8kbps | 12kbps  |        -        |
> > |  **MOS**  |     4.54     | 4.55  |  3.52   |      3.74       |
> >
> > To more accurately represent voice quality and make our results more convincing, we have supplemented the subjective evaluation of our voice change mode experiments by including the latest VC benchmarks, DDDM-VC (AAAI 2024) and NaturalSpeech 3 (FACodec) (ICML 2024). The evaluation involved ten subjects who assessed speech naturalness and speaker similarity on a 5-point mean opinion score (MOS) scale, ranging from 1 (bad) to 5 (excellent).
> >
> > We evaluated six VC systems, focusing on two target timbres (one male and one female).
> > The test set includes two male and three female speakers. This resulted in five conversion pairs for each specific target timbre, leading to a total of 30 converted utterances from the six VC systems evaluated by each subject.
> > Subjects scored naturalness (NMOS) and similarity (SMOS) for 30 converted utterances for the specific target timbre.
> > The subjective results are presented in the table below.
> >
> > Table N-MOS and S-MOS on Male timbre (Correspondence Table 2)
> >
> > |              | N-MOS | S-MOS | Resemblyzer (%) |
> > | :----------: | :---: | :---: | :-------------: |
> > |    VQMIVC    | 3.24  | 2.18  |      66.06      |
> > |   Diff-VC    | 2.94  | 2.60  |      79.74      |
> > |   QuickVC    | **4.05**  | 2.60  |     58.33      |
> > |   DDDM-VC    | 2.00  | 2.60  |      78.19      |
> > |   FACodec    | 2.73  | 2.82  |      81.08           |
> > | VChangeCodec (Ours) | 3.55  | **3.98**  |      **88.07**      |
> >
> > Table N-MOS and S-MOS on Female timbre (Correspondence Table 6)
> >
> > |              |  N-MOS   |  S-MOS   | Resemblyzer (%) |
> > | :----------: | :------: | :------: | :-------------: |
> > |    VQMIVC    |   2.88   |   2.17   |      52.70      |
> > |   Diff-VC    |   3.02   |   2.05   |      70.13      |
> > |   QuickVC    | **4.07** |   2.01   |      47.30      |
> > |   DDDM-VC    |   2.71   |   2.36   |      73.80      |
> > |   FACodec    |   3.35   |   2.61   |      74.11           |
> > | VChangeCodec (Ours)  |  4.00   | **4.38** |    **84.80**    |
> >
> > Based on our findings, we can confidently state that our model attains the highest scores in subjective evaluation for S-MOS and the near-optimal performance in N-MOS. Our method is particularly tailored to specific target timbres, offering less versatility in timbre conversion but superior quality in the conversion of selected timbres. This approach illustrates that the "one timbre, one model" strategy is not only feasible but also robust.

---

> > > ### Author Response · Authors · 2024-11-19
> > > **Response to Reviewer 2ct9 (4/4)**
> > >
> > > Q5: _**VC baselines in Table 2 are not SOTA models, please revise the claim or compare the proposed system with the recent SOTA models like LM-VC, SELFVC, or DDDM-VC.**_
> > >
> > > We appreciate the reviewer's suggestion and have incorporated the latest DDDM-VC and NaturalSpeech 3 (FACodec) system into Table 2 and Table 6 in Q3. The comprehensive comparative results across multiple evaluation dimensions demonstrate the consistent high-quality timbre conversion performance of our VChangeCodec.
> > >
> > > Q6: _**What advantages does VChangeCodec offer over existing state-of-the-art neural speech codecs in terms of parameter efficiency and compression quality?**_
> > >
> > > We must emphasize that we implemented a novel streaming architecture with integrated scalar quantization. The following points will provide a comprehensive understanding of our VChangeCodec.
> > >
> > > - First, We would like to emphasize that our work specifically targets real-time communication (RTC) scenarios, where speech codec efficiency is paramount. Our method has strong RTC features (designing a fully streaming architecture) to fulfill the requirements in low complexity, which is not achievable with codecs like Descript Audio Codec (DAC, Neurips 2024), FACodec (ICML 2024), and SpeechTokenizer (ICLR 2024) in their current designs.
> > > For example, the DAC codec has 75 million parameters, and the FACodec has 1 billion parameters.
> > >
> > > - Secondly, our proposed method can provide prominent performance in both original voice compression quality and voice conversion quality with **a parameter amount of less than 1 million and a low delay of just 40 ms** in Table 4.
> > > We introduce a balanced architecture where the encoder and decoder have comparable parameter counts, as evidenced by our Real-Time Factor (RTF) analysis.
> > > In contrast, existing solutions like DAC and FACodec rely on extremely large decoders for high-quality speech reconstruction.
> > >
> > > -  The DAC codec currently represents the state-of-the-art (SOTA) performance in the neural speech coding field. Our subjective scores indicate that competitive quality is achieved with the lowest delay and parameter quantity in Q4.
> > >
> > > Q7: _**How can VChangeCodec ensure robust performance across various network conditions typical in RTC scenarios?**_
> > >
> > > Recognizing that neural speech coding is inherently data-driven, we incorporated mixed speech segments, including those with noise or music, to enhance the robustness of our scheme. Our test set is strictly an out-of-domain dataset, ensuring it was not exposed to any model during training. In line with the ITU-T P.800 recommendation and adhering to the codec evaluation standard [3], our test set is used to assess all codec models on an identical, unseen test corpus.
> > >
> > > [3] Muller T, Ragot S, Gros L, et al. Speech quality evaluation of neural audio codecs[C]//Proc. Interspeech. 2024.

---

> ### Author Response · Authors · 2024-11-21
> **Request for Follow-Up on Rebuttal**
>
> Dear Reviewer 2ct9:
>
> We would like to thank you once again for taking the time to review our manuscript! Your comments and feedback are highly appreciated.
>
> We have provided a detailed response to every comment. In our **general response**, we have emphasized our innovation and privacy insights. Importantly, we have added **the latest models (DDDM-VC, NaturalSpeech 3) and provided a subjective evaluation**.
>
> We would like to know whether our rebuttals addressed your previous concerns about the more target timbre and SOTA VC models. If they did, **we would greatly value an increase in the score**. Please, let us know if you have any follow-up concerns or comments. We would be happy to clarify those.

---

> > ### Comment · Reviewer_2ct9 · 2024-11-21
> >
> > Thanks for the author's efforts. I appreciate the inclusion of detailed comparisons with the recent codec model and VC models, and I recommend incorporating all these comparisons into the paper.
> >
> > After reading the response, I have one more question. The authors claim that ''the proposed method can provide prominent performance in both original voice compression quality and voice conversion quality with a parameter amount of less than 1 million and a low delay of just 40 ms.'' It makes me confused, about which module or strategy contributed to such impressive results of Table N-MOS and S-MOS on timbre. We can find the SMOS of VChangeCodec is significantly higher than all baseline systems by a large margin (4.xx vs. 2.xx).
> >
> > If the authors would like to claim strong voice conversion ability, please add more details about training data model parameters between the proposed approach and VC baseline systems.

---

> > > ### Author Response · Authors · 2024-11-21
> > >
> > > Dear Reviewer 2ct9:
> > >
> > > We greatly appreciate your insightful and timely feedback.
> > > Please check the updated PDF revision. We have highlighted the revised sentences **in blue font**. For clarity, the new results are temporarily located in the Appendix but consider moving them to the main manuscript.
> > >
> > > We added new baselines DDDM-VC (AAAI, 2024) and NaturalSpeech 3 (FACodec, ICML 2024) in Table 2 and Table 6 in the new PDF revision.
> > > We included the subjective evaluations of the target timbre in Tables 7 and Table 8. (Appendix A.7)
> > >
> > > First, the superior performance stems from VChangeCodec's **high-quality speech reconstruction capabilities**, achieved with a remarkably compact model of only 1 million parameters. We explain the intrinsic benefit of applying the VC task in the token domain, leveraging concepts from information theory. Traditional VC tasks are typically performed through waveform-to-waveform mapping, where each sample is represented in a dynamic range of [-32767, 32768] when digitized with 16-bit depth. In contrast, our method enables a more compact representation in the token domain, where each component is quantized to **2-3 bits**. **The benefit of this 'entropy reduction' is that tokenization**, combined with scalar quantization, allows for a much more compact representation of the data and improves the efficiency of subsequent processing tasks. The effectiveness of this entropy reduction approach is demonstrated through several experiments in our paper, which show significant improvements in both compression efficiency and voice conversion performance.
> > >
> > > Furthermore, our method **adheres to a "one model for one timbre" approach (customized voice change mode in real-time)** for the target RTC use case (please see the detail in general response), utilizing a lightweight model tailored for deployment in mid-range performance smartphones. We have provided additional clarification of "Regarding the differences with zero-shot VC solutions and the limits of target timbre" in **general response**. Our method is more specifically tuned to a certain target timbre. Less generality in timbre conversion, higher quality in selected timbre conversion. In the voice changer mode, we have introduced OpenSmile and established token commitment constraints, along with a plug-and-play causal projection network. We propose to generate target features by converting source speaker utterances via the RVC model, facilitating a near-parallel training database.
> > >
> > > Specifically, for the voice changer,
> > > we use VCTK and AISHELL-3 as the source utterances, they are internally expressive multi-speaker English and Mandarin corpus. All speech utterances are at
> > > a sampling rate of 16 kHz. The parameters of our model are in section 3.2 and “Model and training recipe”.
> > > For all the VC baseline systems, we use the official open-source codes and pre-trained model to realize zero-shot VC inference. They all keep the default parameter configuration. All input speech utterances are at a sampling rate of 16 kHz for alignment.
> > > For further clarification, we have added a detailed description in section 4.1 and Appendix A.5 - Experimental details.
> > >
> > > Let us know if you have any follow-up concerns or comments. We would be happy to clarify those.

---

> ### Author Response · Authors · 2024-11-22
> **Response to Reviewer 2ct9's question**
>
> Dear Reviewer 2ct9,
>
> Let us explain "why does VChangeCodec work well?".
> Here, we will present an intuitive explanation.
>
> First, high-quality speech reconstruction is achieved using a pre-trained codec with less than 1 million parameters. Let $X$ denote a source segment and $T$ is a target segment.  $\hat{z}_x$ denotes the token of source segment. So the transformation chain is: $X$ --> $z_x$ --> $\hat{z}_x$ --> $\hat{X}$ ~= $X$.
> The objective is to minimize the reconstruction loss in the waveform domain:
>
> $$ argmin || \hat{X}  - X|| \tag{1}$$
>
>
> $$ argmin || \hat{T}  - T|| \tag{2}$$
>
>
> In the token domain (dimension $N =84$), we aim to minimize the difference between the $ \hat{z}_x $ and $ \hat{z}_t $, to achieve the best reconstruction of $T$ after the decoding process.
>
> The token contains all content and speaker information $u_s$ for perfect reconstruction. To realize perfect voice conversion, we need to keep content information and modify the target speaker information $u_t$. For specific target speakers, we refer to the SOTA voice conversion model to generate nearly parallel data from source utterances, which keeps the content information.
>
> We design a causal projection network to learn this timbre adaption $f$. We introduce a token commitment loss to ensure token-level minimization.
>
> $$argmin || f(\hat{z}_x) - \hat{z}_t|| \tag{3}$$
>
> This leads to:
>
> $$ argmin || \hat{z}_{t} - \hat{T} || \tag{4}$$
>
> Combined with equation (2), we can obtain the high-quality target speech reconstruction.
> Moreover, we utilize the acoustic features extracted by openSMILE, which can represent speaker identity (timbre information) and capture the subtle emotional variations in speech.
>
> In an ideal case, the output token $\hat{z}_t$ does not contain any information about the source speaker $u_s$. This elegant design enables VChangeCodec to achieve high-quality VC with less than 1 million parameters, while the lowest latency and almost causal  VC (AC-VC) in our knowledge requires at least 2 million parameters.
>
> Ronssin D, Cernak M. AC-VC: non-parallel low latency phonetic posteriorgrams based voice conversion[C]//2021 IEEE Automatic Speech Recognition and Understanding Workshop (ASRU). IEEE, 2021: 710-716.

---

> > ### Comment · Reviewer_2ct9 · 2024-11-24
> >
> > Thanks for the response. It addressed some of my concerns, but I think the comparison of voice conversion baselines is unfair. All VC baseline systems can achieve any-to-any conversion, while the proposed only supports any-to-one or one-to-one. On the other hand, the parallel training data is also undesire, since baseline systems are trained using unparallel data by a disentanglement pattern.
> >
> > Therefore, I intend to keep my score unchanged.

---

> > > ### Author Response · Authors · 2024-12-02
> > > **Request for Follow-Up on Rebuttal**
> > >
> > > Dear Reviewer  2ct9,
> > >
> > > We kindly hope **you could find some time to review our latest responses and new experiments by retraining selected baseline VC systems with our target timbre dataset**.
> > >
> > > Given the extended discussion period, there is still an opportunity for questions and feedback. We hope to fully address your concerns.
> > >
> > > We understand that your time is valuable and you may be busy with other things.
> > > However, your insights would be extremely valuable for improving our work.
> > > We greatly appreciate your consideration.

---

> ### Author Response · Authors · 2024-11-24
> **Response to Reviewer 2ct9's new question**
>
> Dear Reviewer 2ct9,
>
> We greatly appreciate your follow-up concerns and timely feedback.
> To comprehensively evaluate our model's competitiveness, we conducted fair experiments by **retraining selected baseline VC systems with our target timbre dataset** (Male timbre, correspondence Table 2). While Diff-VC and FACodec were excluded due to unavailable f0/spectrogram extraction scripts and training procedures respectively, we focused on three advanced systems: VQMIVC, QuickVC, and the recent DDDM-VC.
>
> Retraining details are as follows:
> VQMIVC: Trained from scratch for 500 epochs using extracted f0 and spectral features from target male timbre data.
> QuickVC: Fine-tuned for 3200k steps on their provided base model at 1200k steps  (due to significant training time requirements).
> DDDM-VC: Trained from scratch for 200k steps with offline-extracted f0.
>
> The objective evaluation results are presented in the table below. Bold indicates the best result, and italics indicate the second-best result.
> VQMIVC showed decreased performance, likely due to the need for retraining a dataset-specific vocoder, which requires more complexity. Considering the limited time, we did not retrain the vocoder.
> QuickVC demonstrated significant improvement in timbre similarity, though still not reaching our performance.
> DDDM-VC showed improvements across all objective metrics.
> Nevertheless, VChangeCodec **maintains competitive timbre adaptation capabilities and voice quality**. Crucially, our system achieves these results with **full streaming capability and a lightweight architecture of less than 1 million parameters**. We included these experiments of **the retrained target timbre in Tables 10 (Appendix A.9)**.
>
> Moreover, we intentionally designed our system for any-to-one voice conversion, as this matches **our target application scenario where the target timbre is exclusively maintained by mobile network operators**. By deliberately not implementing any-to-any conversion capabilities, we mitigate potential privacy risks, as regular users cannot modify or access unauthorized target timbres.
>
> We have substantially enhanced our paper with comprehensive evaluations and fair experimental settings. We would like to emphasize that the primary contribution of our work **lies in the novel speech codec architecture with VC features**.  The evaluation against zero-shot voice conversion models has already substantially demonstrated the competitive voice quality capabilities of our VChangeCodec.
> **We believe that this comprehensive experiment will alleviate your concerns.**
>
> Table Comparison with the retrained/finetuned SOTA VC methods on Male timbre (correspondence Table 2)
>
> |                     |   SIG    |   BAK    |   OVRL   |   MOS    |   MCD    |    WER    |    CER    | Resemblyzer |
> | :-----------------: | :------: | :------: | :------: | :------: | :------: | :-------: | :-------: | :---------: |
> |       VQMIVC        | **3.46** |   3.82   |   3.03   |   2.95   |   6.58   |  118.96%  |  89.71%   |   56.61%    |
> |       QuickVC       |  _3.38_  | **4.11** | **3.16** | **3.74** |  _6.31_  | **9.07%** | **4.96%** |  _87.57%_   |
> |       DDDM-VC       |   2.21   |   3.28   |   1.96   |   3.34   |   6.73   |  29.49%   |  13.97%   |   83.00%    |
> | VChangeCodec (Ours) |   3.35   | **4.11** |  _3.11_  |  _3.71_  | **5.76** | _16.19%_  |  _7.67%_  | **88.07%**  |
> |  Oracles (Target)   |   3.29   |   4.04   |   3.06   |   3.84   |          |           |           |    100%     |
>
>
> We welcome any further questions you may have and would be more than happy to provide prompt responses. We sincerely hope you will consider **re-evaluating the merits of our work**.

---

> ### Author Response · Authors · 2024-11-25
> **New experiments in response to reviewer 2ct9**
>
> Dear Reviewer 2ct9:
>
> We value active discussions with you. I hope you will take the time to read **our new experiments using retrained versions of the other state-of-the-art methods** above. This is a fair experiment to support our previous conclusions. We sincerely hope you will **consider re-evaluating the merits of our work**.
>
> In general, our approach is a completely new approach targeting the real-time communication (RTC) scenario with distinctive technical requirements and design, in which the technical problems and requirements are quite different from generalized voice conversion methods.
>
> First, the key difference between our approach and traditional voice conversion (VC) models lies in their target audiences and operational contexts. **Our lightweight neural speech codec is embedded within RTC systems, where the encoder and decoder are immutable and maintained by mobile network operators**.
> Consequently, it is not reasonable to compare with VC models on the aspect of the performance in VC only but to compare the performance and the merit of our method, comprehensively.
> In our opinion, current VC models are hard to fulfill the RTC requirements. The transmission process of VC in RTC is shown in Figure 1 (a), placing a lightweight VC model either as a pre-processor or post-processor of the codec (e.g., VC-->Codec or Codec-->VC) would inevitably introduce double latency for both VC processing and compression.
> For example, the lightest AC-VC model [1] exhibits an algorithmic **delay of 57.5 ms and at least two million parameters** when running on a CPU infrastructure. Considering the additional 10 ms delay of the LPCNet pitch predictor in this system, and the 40 ms delay of the speech codec, the delay will reach **a cumulative latency of over 107.5 ms**.
> This significantly exceeds the acceptable latency threshold for RTC requirements.
>
> Secondly, our approach fundamentally differs from this lightweight VC model in methodology and efficiency. Instead of treating VC as a separate process, we integrate timbre adaptation directly within the codec transmission pipeline in Figure 1 (b). Our method maintains **just a 40 ms algorithmic delay without introducing any additional latency and less than one million parameters**.
> **This novel framework represents a paradigm shift from traditional cascaded VC-codec systems to an integrated, efficient solution that achieves timbre adaptation and compression in a single unified pipeline.**
>
> Thirdly, we have carefully **considered privacy issues for applications in RTC services**. Our VC module is integrated into the speech communication system, prohibiting users from accessing or altering internal configurations. The embedded speaker representation is injected at the sending end and maintained by the operators, with designated timbres being pre-defined and inaccessible to ordinary users. A potential use case is expected the user can only download the related binary file after subscribing to a specific target timbre from the operator. Consequently, the user cannot arbitrarily change the source voice to any target timbre to minimize the deepfake risk to operators.
>
> We welcome any further questions you may have and would be happy to provide prompt responses.

---

> ### Author Response · Authors · 2024-11-27
> **Request for Reviewer 2ct9: Fair Experiments & PDF Update**
>
> Dear Reviewer 2ct9:
>
> Thank you again for your valuable suggestions about the latest baselines and guidance throughout this review process. I hope you will take the time to read **our new experiments using retrained SOTA VC methods (Please refer to the previous response box)**. This is a fair experiment that fully addresses your concerns and further supports our previous conclusions.
>
> Importantly, we have carefully integrated all experiments into the main body of the paper. We sincerely invite you to **take some time to read our revised PDF**. You should be able to see it by clicking the top-right PDF button. We used **the blue font to highlight the new content**. We revised the manuscript according to your suggestions as follows:
>
> - We have added subjective evaluations and a discussion of speech codec (in lines 385-390).
>
> - We have added subjective experiments in Table 3 (in lines 432-447) and Table 8 (in lines 912-929).
>
> - Additionally, we have added the experiment on retrained VC models in Table 4 and the corresponding analysis including more details about model parameters of VC baseline systems (in lines 449-465).
>
> We sincerely hope you will consider re-evaluating the merits of our work. We hope we addressed all your concerns, and in this case would be happy if you could consider increasing the score, and if not we are more than willing and happy to engage in a discussion with you to answer further questions.

---

> ### Author Response · Authors · 2024-11-29
> **We would like to follow up on our rebuttal**
>
> Dear Reviewer 2ct9,
>
> We sincerely thank you once again for your effort in reviewing our manuscripts. We kindly hope **you could find some time to review our latest responses**.
>
> We have worked to address your concern through additional experiments and clarifications. We outline them below:
>
> - **[Fair comparison with our target timbre dataset]** As you requested, we conducted additional experiments by comparing our VChangeCodec with SOTA VC models. **Using the same training dataset of the selected target timbre**, we ensured fairness and consistency throughout the evaluations. Such experiments can be viewed as an equivalent comparison to any-to-one models.
>
> - **[SOTA performance in all evaluations]** We have demonstrated the superior performance of our VChangeCodec through extensive experiments in Table 4.
>
> - **[Key difference between our approach and traditional VC models]** Our lightweight neural speech codec is embedded within RTC systems, where the encoder and decoder are immutable and maintained by mobile network operators. Consequently, it is not reasonable to compare with VC models on the aspect of the performance in VC only but to compare the performance and the merit of our method, comprehensively.
>
> - **[The updated PDF manuscript]** We have carefully integrated all experiments into the main body of the paper.
>
> We hope these substantial improvements will be considered in the final evaluation of our paper. We remain open to further discussion to enhance our work and welcome any additional questions you may have.
>
> Thank you once again for your constructive feedback.
>
> Sincerely,
>
> Authors.

---

### Official Review · Reviewer_Y2Se · 2024-11-02

**Soundness:** 3
**Presentation:** 2
**Contribution:** 2
**Rating:** 6
**Confidence:** 3

**Summary:**

This paper introduces a new neural speech codec designed to integrate voice-changing capabilities directly into the codec itself. This integration allows for switching between original and customized voice modes in real-time, making it efficient for bandwidth-constrained environments. The proposed method leverages a causal projection network within its encoding module to adapt the timbre of the transmitted voice at the token level, achieving latency of 40ms and requiring fewer than 1 million parameters. This makes it ideal for real-time communication (RTC) scenarios such as online conferencing.

This paper highlights the limitations of existing neural speech codecs (NSCs) and voice conversion (VC) systems, which typically operate separately and introduce additional latency. VChangeCodec addresses these issues by combining speech compression and voice conversion into a single, integrated framework. The codec uses scalar quantization to reduce complexity and maintain high fidelity at lower bitrates. Comprehensive evaluations, including subjective listening tests and objective performance assessments, demonstrate that proposed method results in timbre adaptation capabilities, providing a flexible solution for RTC systems.

Additionally, the paper discusses the technical details of codec's architecture, including its encoder, quantization, and decoder components, as well as the training strategy involving multiple loss functions to ensure high-quality speech reconstruction and timbre adaptation. The authors emphasize the operator-oriented deployment of proposed technique, which minimizes privacy risks by restricting user access to pre-defined timbres. The results of extensive experiments and ablation studies show the effectiveness of VChangeCodec, making it a possible approach for enhancing real-time communication with built-in voice-changing features.

**Strengths:**

Well written paper with an interesting proposition to carry out voice conversion as part of the codec. The background and methods section is meticulously written and explained nicely. The scalar quantization is a know technique from past which is now getting revived in the context of neural network. The authors have evaluated their models in objective and subjective metrics showing either improvement in performance or matching state-of-the-art codecs and VC models.

**Weaknesses:**

The main weakness is the motivation which I fail to understand at this point. If a light-weight Voice conversion model can be used a post-processor or a pre-processing module after/before codec, then what additional advantage does this framework brings. Second, apart from the combination of Codec+VC module and scalar quantization trick, there is no axis of novelty in this paper. Additionally, in the experiment section, evaluation of speaker similarity through SMOS would be more convincing than resemblyzer model. Finally, the usage of WER and CER (by Whisper model) does not suggest greater intelligibility as they inherently make use of a language model to correct pronunciation mistakes.

**Questions:**

None

---

> ### Author Response · Authors · 2024-11-20
> **Response to Reviewer Y2Se (1/3)**
>
> Dear Reviewer Y2Se,
>
> We sincerely appreciate your positive feedback and are pleased to see your deep understanding of our paper's content and objectives. We will respond to your questions point by point.
>
>
> > Q1: _**The main weakness is the motivation which I fail to understand at this point. If a light-weight Voice conversion model can be used a post-processor or a pre-processing module after/before codec, then what additional advantage does this framework brings.**_
>
> - First, placing a lightweight VC model either as a pre-processor or post-processor of the codec (e.g., VC-->Codec or Codec-->VC) would inevitably introduce double latency for both VC processing and compression.
> For example, the lightest AC-VC model to our knowledge [1], when run on a CPU, exhibits an algorithmic delay of 57.5 ms. Considering the additional 10 ms delay of the LPCNet pitch predictor in this system, and the 40 ms delay of the speech codec, the delay will reach **a cumulative latency of over 107.5 ms**.
> This significantly exceeds the acceptable latency threshold for real-time communication requirements.
>
> - Secondly, our approach fundamentally differs from this lightweight VC model in methodology and efficiency. Instead of treating VC as a separate process, we integrate timbre adaptation directly within the codec transmission pipeline. By incorporating a causal projection network into the existing speech codec architecture, our method maintains **just a 40 ms algorithmic delay without introducing any additional latency**.
> Furthermore, while conventional SOTA VC methods operate on waveform-to-waveform mapping, which is computationally intensive and impractical for mobile devices, our approach processes low-dimensional tokens, significantly reducing computational complexity. This novel framework represents a paradigm shift from traditional cascaded VC-codec systems to an integrated, efficient solution that achieves both timbre adaptation and compression in a single unified pipeline.
>
> [1] Ronssin D, Cernak M. AC-VC: non-parallel low latency phonetic posteriorgrams based voice conversion[C]//2021 IEEE Automatic Speech Recognition and Understanding Workshop (ASRU). IEEE, 2021: 710-716.

---

> > ### Author Response · Authors · 2024-11-20
> > **Response to Reviewer Y2Se (2/3)**
> >
> > > Q2: _**Apart from the combination of Codec+VC module and scalar quantization trick, there is no axis of novelty in this paper.**_
> >
> > We thank you for recognizing our main contribution lies in proposing an innovative method with a solid embodiment and pipeline of the speech codec with new VC features. We must emphasize that we implemented **a novel streaming architecture with integrated scalar quantization**. In the voice changer mode, we have introduced OpenSmile and established token commitment constraints, along with a plug-and-play causal projection network. The following points will provide a comprehensive explanation of our innovation.
> >
> > - First, we would like to emphasize that our work specifically targets real-time communication (RTC) scenarios, where the efficiency of speech codecs is critical. Our method has strong RTC features (designing a fully streaming architecture) to fulfill the requirements in low complexity， which is not achievable with codecs like Descript Audio Codec (DAC, Neurips 2024), FACodec (ICML 2024), and SpeechTokenizer (ICLR 2024) in their current designs.
> > For example, the DAC codec has 75 million parameters, and the FACodec has 1 billion parameters. According to the latest codec evaluation [2], the performance gains obtained by DAC et al. may come with increased complexity.
> >
> > - Secondly, our proposed method can provide prominent performance in both original voice compression quality and voice conversion quality with **a parameter amount of less than 1 million and a low delay of just 40 ms**.
> > We present a balanced architecture in which the encoder and decoder have comparable parameter counts, as demonstrated by our Real-Time Factor (RTF) analysis in Table 5.
> > In contrast, existing solutions like DAC and FACodec rely on extremely large decoders for high-quality speech reconstruction.
> >
> > - **Privacy considerations** are a cornerstone of our contribution, and to the best of our knowledge, this paper is the first to address this specific aspect.
> >
> > In this section, we explain the intrinsic benefit of applying the VC task in the token domain, leveraging concepts from information theory. Traditional VC tasks are typically performed through waveform-to-waveform mapping, where each sample is represented in a dynamic range of [-32767, 32768] when digitized with 16-bit depth. In contrast, our method enables a more compact representation in the token domain, where each component is quantized to **2-3 bits**. The benefit of this 'entropy reduction' is that tokenization, combined with scalar quantization, allows for a much more compact representation of the data and improves the efficiency of subsequent processing tasks. The effectiveness of this entropy reduction approach is demonstrated through several experiments in our paper, which show significant improvements in both compression efficiency and voice conversion performance.
> >
> > For further clarification, you may refer to our detailed explanation of "Emphasizing our innovations in neural codec field and operator-oriented backgrounds" in the **general response**. We also have elaborated on the innovation in the neural codec field and operator-oriented backgrounds.
> >
> > [2] Muller T, Ragot S, Gros L, et al. Speech quality evaluation of neural audio codecs[C]//Proc. Interspeech. 2024.

---

> > > ### Author Response · Authors · 2024-11-20
> > > **Response to Reviewer Y2Se (3/3)**
> > >
> > > > Q3: _**In the experiment section, evaluation of speaker similarity through SMOS would be more convincing than resemblyzer model.**_
> > >
> > > We appreciate the your suggestion about the importance of SMOS evaluation for speaker similarity.
> > >
> > > To enhance the credibility of our results and provide a more precise assessment of voice quality, we have supplemented the subjective evaluation of our voice change mode experiments by including the latest VC benchmarks, **DDDM-VC (AAAI 2024) and FACodec (ICML 2024)**. The evaluation involved ten subjects who assessed speech naturalness and speaker similarity on a 5-point mean opinion score (MOS) scale, ranging from 1 (bad) to 5 (excellent).
> > >
> > > We evaluated six VC systems, focusing on two target timbres (one male and one female).
> > > The test set includes two male and three female speakers. This resulted in five conversion pairs for each specific target timbre, leading to a total of 30 converted utterances (five pairs * six systems) from the six VC systems evaluated by each subject.
> > > Subjects scored naturalness (N-MOS) and similarity (S-MOS) for 30 converted utterances for the specific target timbre.
> > > The subjective results are presented in the table below.
> > >
> > > Table N-MOS and S-MOS on Male timbre (Correspondence Table 2)
> > >
> > > |              | N-MOS | S-MOS | Resemblyzer (%) |
> > > | :----------: | :---: | :---: | :-------------: |
> > > |    VQMIVC    | 3.24  | 2.18  |      66.06      |
> > > |   Diff-VC    | 2.94  | 2.60  |      79.74      |
> > > |   QuickVC    | **4.05**  | 2.60  |     58.33      |
> > > |   DDDM-VC    | 2.00  | 2.60  |      78.19      |
> > > |   FACodec    | 2.73  | 2.82  |      81.08           |
> > > | VChangeCodec (Ours) | 3.55  | **3.98**  |      **88.07**      |
> > >
> > > Table N-MOS and S-MOS on Female timbre (Correspondence Table 6)
> > >
> > > |              |  N-MOS   |  S-MOS   | Resemblyzer (%) |
> > > | :----------: | :------: | :------: | :-------------: |
> > > |    VQMIVC    |   2.88   |   2.17   |      52.70      |
> > > |   Diff-VC    |   3.02   |   2.05   |      70.13      |
> > > |   QuickVC    | **4.07** |   2.01   |      47.30      |
> > > |   DDDM-VC    |   2.71   |   2.36   |      73.80      |
> > > |   FACodec    |   3.35   |   2.61   |      74.11           |
> > > | VChangeCodec (Ours)  |  4.00   | **4.38** |    **84.80**    |
> > >
> > > The experimental results demonstrate that our model achieves superior performance in S-MOS and attains near-optimal performance in N-MOS.
> > > The subjective evaluation results not only align with our objective evaluation but demonstrate even stronger performance, particularly in speaker similarity (SMOS). This consistent performance across both objective and subjective measures reinforces the robustness of our approach.
> > > It illustrates that our strategy is highly effective, delivering high-quality voice conversion for targeted applications.
> > >
> > > We have provided additional experimental results in **general response**, along with a new demonstration page. **We would greatly appreciate your time in comparing the subjective quality at the following link**:
> > >
> > > https://anonymous666-speech.github.io/Demo-VChangeCodec/
> > >
> > >
> > > > Q4: _**The usage of WER and CER (by Whisper model) does not suggest greater intelligibility as they inherently make use of a language model to correct pronunciation mistakes.**_
> > >
> > > We acknowledge the reviewer's insightful observation about the potential limitations of using WER/CER metrics derived from Whisper, given its internal language model's error correction capabilities.
> > >
> > > However, we believe this metric still provides valuable insights for several reasons:
> > >
> > > - We conducted comparative evaluations where all systems were assessed under identical conditions using the same "small" Whisper model, **ensuring a fair comparison baseline**.
> > >
> > > - While Whisper's language model may correct some pronunciation errors, significant mispronunciations or distortions would still likely result in recognition errors, making WER/CER useful as relative indicators of speech quality.
> > > Most importantly, we have added **our subjective evaluation results**, where human listeners directly assessed naturalness and speaker similarity, providing a more comprehensive evaluation of our system's performance.
> > >
> > > - Furthermore, we also evaluated our system using additional objective metrics such as Mel Cepstral Distortion (MCD), which are independent of any language model effects.
> > >
> > > - Finally, our choice of the Whisper model for intelligibility assessment is well-established in the field. It serves as the standard evaluation metric in the Codec-SUPERB competition [3], and the same Whisper en-medium model has been adopted by recent influential work such as SpeechTokenizer (ICLR 2024) [4]. This validates the reliability of Whisper-based metrics for speech intelligibility evaluation.
> > >
> > > [3] https://github.com/voidful/Codec-SUPERB
> > >
> > > [4] Zhang X, Zhang D, Li S, et al. Speechtokenizer: Unified speech tokenizer for speech language models[C]//The Twelfth International Conference on Learning Representations. 2024.

---

> ### Author Response · Authors · 2024-11-22
> **Request for Follow-Up on Rebuttal**
>
> Dear Reviewer Y2Se:
>
> We would like to thank you once again for taking the time to review our manuscript! Your comments and feedback are highly appreciated.
>
> We have provided a detailed response to every comment. In our **general response**, we again emphasized the novelty of our approach. We added comprehensive subjective evaluations in Table 7 and 8 (Appendix A.7) and the reference (Appendix A.10).
>
> We would like to know whether our rebuttals and new PDF revision addressed your previous concerns. If they did, we kindly ask you to consider updating your score to reflect the improvements made. If there are any questions, we would be happy to provide additional details.

---

> > ### Comment · Reviewer_Y2Se · 2024-11-25
> > **Followin up on the rebuttal**
> >
> > I thank the authors for providing details on the latency, motivation, and adding subjective listening results in their response. These details are expected to be present in the paper for the sake of completeness and reproducibility. I am keeping my score unchanged citing these reasons.

---

> ### Author Response · Authors · 2024-11-26
> **Response to reviewer's feedback about a new PDF**
>
> Dear Reviewer Y2Se:
>
> We appreciate your insightful response and constructive suggestions. Considering the page limit, we have tried our best to reasonably incorporate new content into the main body of the paper. **We sincerely invite you to take the time to check the latest PDF revision.** We used **the blue font to highlight the new content**. We believe that **the new version can meet the requirements of the ICLR conference, with complete experiments that are reproducible.**
>
> - Specifically, we have added the discussion about latency and motivation in the introduction (in lines 83-87 and 92-94).
> - We have added subjective experiments (including the latest baseline system) in Table 3 (in lines 432-447) and Table 8 (in lines 912-929).
> - Additionally, we have provided a more detailed experimental discussion and experimental details (in lines 426-431 and 864-886).
> - We have also incorporated the suggestions of the other reviewers into the paper.
>
> All of these modifications make our article more complete and reproducible.
> Therefore, these improvements significantly enhanced the manuscript's academic depth and comprehensiveness of our manuscript.
>
> Your suggestions have inspired us to further improve the quality of our article. **It is because of your support that our work can progress and continuously improve. If the new PDF version resolve your concerns, we would greatly value an increase in the score.** If you have any new questions, we'll be happy to continue addressing them.

---

> ### Author Response · Authors · 2024-11-29
> **We would like to follow up on our rebuttal**
>
> Dear Reviewer Y2Se,
>
> We sincerely thank you once again for your effort in reviewing our manuscripts. We kindly hope **you could find some time to review our latest responses**.
>
> As you requested,  we have carefully integrated all details on latency, motivation, and subjective evaluation results into the main body of the paper.
> We hope these substantial improvements will be considered in the final evaluation of our paper. We remain open to further discussion to enhance our work.
>
> Thank you once again for your constructive feedback.
>
> Sincerely,
>
> Authors.

---

> ### Author Response · Authors · 2024-12-02
> **Friendly reminder to review the latest response**
>
> Dear Reviewer Y2Se,
>
> As the discussion phase is approaching the end, we sincerely hope **you could find some time to review our latest responses**. We hope to fully address your concerns.
>
> We understand that your time is valuable and you may be busy with other things.
> However, your insights would be extremely valuable for improving our work.
> We greatly appreciate your consideration.

---

### Official Review · Reviewer_UYJc · 2024-11-04

**Soundness:** 3
**Presentation:** 2
**Contribution:** 3
**Rating:** 6
**Confidence:** 4

**Summary:**

The paper proposes a real-time voice codec with an integrated vocal identity conversion module. It first trains the original mode of the codec and validates that the performance is approximately similar to other state-of-the-art codecs. Then the authors propose a voice changer module that is a causal convolutional network that is conditioned on the target speaker using a selected set of features provided by the OpenSmile software. To be able to construct target features a parallel training database is required. To this end, the paper proposes to create the target phrases synthetically by means of converting the input phrase of the source speaker using the RVC model.
The voice changer mode is evaluated by means of comparing it to results of a few selected algorithms from the literature.

**Strengths:**

- Very original idea to combine a voice codec with an integrated identity conversion component,
- real-time voice identity conversion with low latency is a very ambitious target
- using the existing features of the OpenSmile software as a means to condition the target speaker identity is new (at least to me), and looks potentially very interesting.

**Weaknesses:**

- Not all final parameters are explicitly described. I wonder about the R parameter. The description in the appendix says in line 795:

*Given the target bitrate r, the dimension of latent feature N, the theoretical bitrate in each frame is computed as −1 ∗ N ∗ log2( 1/(2∗R+1)).*

I suggest explicitly saying here what R is supposed to be in this equation and how you ensure the desired target bitrate is achieved.

- Some of the claims are rather misleading. For example, authors state that in original mode their codec works on party with the DAC and Encodec. These two are trained as codecs for arbitrary signals (music included), while the proposed codec is for speech only. The version of Encodec is not clearly specified (authors should add the information about, which model is used in their experiments), but both DAC and Encodec work on signals with higher sample rate. That your codec outperforms general-purpose codecs training to compress signals with a larger sample rate is a positive point, but is not that astonishing either.  A somewhat similar comment applies to Lyra2, which supports arbitrary languages (probably trained in 24kHz but I am not 100% sure about that).  I suggest adding an explicit description of the differences in training data and sample rates when discussing the performance comparisons.
- The description of the data sources in 4.1 is a bit confusing. You write:
  *The clean speech is from LibriTTS (Zen et al., 2019), DNS Challenge (Reddy et al., 2020). The mixed
speech is generated by combining clean speech and background interference (e.g., noise), including
DNS Challenge, MIR-1K (Hsu & Jang, 2009) and FMA (Defferrard et al., 2016).*
A few details should be added:
  - which part of the DNS challenge data did you use? There are singing and expressive datasets mentioned in the Reddy paper.
  - Why do we need mixed speech? This is not mentioned anywhere in the paper. How do you perform the mixes with respect to the balance between background (noise) and foreground (voice)
  - Nothing is said about the data that is used for validation. Please add a description of the train/validation/test split.

- While perceptual evaluation is performed for the codec in original mode, neither DAC nor Encodec are part of the comparison. To give a complete picture of the performance of your codec compared with these two, it would be preferable to have them added to the perceptual evaluation. If you cannot add them then please explain why you think it is not possible or not needed.
- The fact that the training is performed using synthetically generated parallel training data is quite hidden, it should be mentioned earlier in the description, for example in the introduction.
- Evaluation of the voice change mode is very weak, no subjective evaluation is provided, and all the baseline models are trained on the VCTK dataset. Even if the models support zero-shot conversion it is clearly unfair to compare these models to your model that - as far as I see - is trained particularly on the target speakers.
The fact that Resemblyzer similarity is low for QuikVC is clearly due to the fact that you operate it out of its context.
- Resemblyzer is a weak alternative to perceptual evaluation.

- English language should be improved. Notably, *Casual projection network* should be renamed into *Causal projection network*. When you say your model *achieves superior latency* it would mean it has a larger latency. you probably want to say your latency is lower than that of the other methods.

**Questions:**

Please explain better how you perform the bit rate calculation in A.3 and specify the quantizer levels R that you use in your experiments.

You should discuss the fact that your model is trained on a parallel database of the target speaker, while all other VC models are operated in zero short mode. The results of your model are difficult to judge.

Why are neither DAC nor Encodec part of the perceptual evaluation in the original mode?

If I understand correctly, your quantization model is exactly the same as the one used in SimpleSpeech and SimpleSpeech 2.
https://arxiv.org/pdf/2408.13893 - if this is correct you should cite one of them.

Remark after the discussion phase:

All these questions have been answered and taken care of by the authors.

---

> ### Author Response · Authors · 2024-11-20
> **Response to Reviewer UYJc (1/4)**
>
> Dear Reviewer UYJc,
>
> We truly value your positive feedback and are glad to see your constructive suggestions. We will address each of your questions in turn.
>
> > Q1: _**Not all final parameters are explicitly described. I wonder about the R parameter.**_
>
> We appreciate the reviewer's request for clarification regarding the parameter R and the mechanism for achieving the target bit rate. We will provide a detailed explanation in our revised manuscript.
>
> Entropy is the same as the common sense in Information Theory, i.e., the average number of bits for coding each symbol in the codebook. For example, the codebook is composed of $\{-1.0, -0.5, 0, 0.5, 1.0\}$ with codebook size 5. The Shannon entropy H(X), is defined by $ H(X) = -\sum p(x_i) * \log_2 p(x_i)$, where the $p(x_i)$ is the possibility of the $i^{th}$ symbol in the above codebook.
>
> In this paper, the value of R is 2. For the 84-dimensional codec model, using a codebook size of 5, the bit rate calculation using Shannon's formula is $−84 × log_2​(1/5)×50/1000=9.75$ kbps. Since we are considering a uniform distribution where entropy is maximized, the actual bit rate will be lower, at 9.5 kbps.
> For the 56-dimensional model, also with a codebook size of 5, according to Shannon's formula, $−56×log_2​(1/5)×50/1000=6.5$ kbps, and the actual bit rate in practice is 6 kbps.
>
> We have supplemented the parameter of the relevant bit rates in the Appendix (A.3).
>
>
> > Q2: _**I suggest adding an explicit description of the differences in training data and sample rates when discussing the performance comparisons.**_
>
> We appreciate the reviewer's suggestion regarding the configuration of baseline codecs (DAC and Encodec).
> We have added comprehensive details about the baseline codecs, including sampling rates and specific model versions. Due to page limitations, these detailed descriptions have been included in the appendix.
>
> First, our VChangeCodec is **a lightweight, low-latency speech codec tailored for mobile network operators**.
> We target deploying the codec through an operator rather than via peer-to-peer communication (as detailed in lines 96-101). We focus on Real-Time Communication (RTC) applications such as online meetings and voice calls, where speech is the primary content. Furthermore, we also add some mixed speech (including speech with background noise and speech with music) for robustness.
> Moreover, DAC and Encodec are trained on extensive audio (speech, music) datasets, which generally leads to better generalization on unseen test data. In this context, our approach achieves comparable quality when evaluated on speech datasets, providing a fair basis for comparison within our target domain.
>
> Secondly, for RTC service, **16kHz is the standard sampling rate**, as the voice is the primary component in-service. Higher sampling rates are typically for audio (including music), but we focus on speech, operating at a sampling rate of 16kHz. For fair comparison in our experiments, systems (DAC, Encodec, Lyra2) with higher sampling rates were downsampled to 16kHz to ensure consistent evaluation conditions.
> Specifically, we used the official Encodec versions [1] for 12 kbps (n_q =16) and 24 kbps (n_q=32), with the model's sampling rate at 24kHz. We downsampled the original test audio from 48kHz to 24kHz for input into the encoder model and downsampled the output speech to 16kHz to compare the quality at the same sampling rate.
> Similarly, for DAC [2], we used the official configured at 16kHz, the default 8kbps model for inference. For Lyra2 [3], we conducted evaluations using the official 16kHz, default 6kbps and 9.2kbps model for inference.
>
> Finally, our model is comparable to DAC (**the number of parameters is reduced by 70x**) in objective speech quality and completely surpasses Encodec at 24kbps.
>
> [1] https://github.com/facebookresearch/encodec
>
> [2] https://github.com/descriptinc/descript-audio-codec/tree/main
>
> [3] https://github.com/google/lyra

---

> ### Author Response · Authors · 2024-11-20
> **Response to Reviewer UYJc (2/4)**
>
> > Q3: _**Add the detailed description of the data sources.**_
>
> - Regarding the training set in original voice mode, we use DNS challenge 2020 dataset and the LibriTTS dataset as the speech part of the training set. As introduced in the last question (Q2), extra mixed speech utterances (mixed with noise or music) are also included, in which the noise clips are from the DNS challenge, and music clips are from MIR-1k and FMA. This configuration is designed for actual RTC scenarios, including pure voice communications, or voice with background interference (e.g. office noise, background music playback, etc).
> We randomly selected noise crops and adjusted the mixing gain of the noise component using SNR, and the target SNR is 15 dB.
> $SNR = 10 log10 (S^2 / (kN)^2)$, S denotes clean signal energy and N is noisy signal energy.
>
> - Recognizing that neural speech coding is inherently data-driven, we incorporated mixed speech segments, including those with noise or music, to enhance the robustness of our scheme.
>
> - For all training data, we randomly sample 98% of the dataset for train, 1% for valid and 1% for test.
> Finally, the unseen test set is strictly an out-of-domain dataset, ensuring it was not exposed to any model during training. In line with the ITU-T P.800 recommendation and adhering to the codec evaluation standard [4], our test set is used to assess all codec models on an identical, unseen test corpus.
>
> [4] Muller T, Ragot S, Gros L, et al. Speech quality evaluation of neural audio codecs[C]//Proc. Interspeech. 2024.
>
> > Q4: _**Give a complete picture of the performance of your codec compared with DAC and Encodec.**_
>
> We appreciate the reviewers' inquiries and understand this question.
>
> Let us clarify why DAC and Encodec were not included in Figure 3. Specifically, our method addresses Real-Time Communication (RTC) requirements through a fully streaming architecture with low computational complexity. Current implementations of codecs like Descript Audio Codec (DAC, Neurips 2024) and Encodec (2022) face significant limitations in meeting these RTC requirements.
> Specifically, DAC's architecture, with its 75 million parameters (**VChangeCodec only needs 1 million parameters**), does not support streaming inference, making it unsuitable for real-time applications. While Encodec does offer streaming capabilities, its processing speed (measured in Real-Time Factor, RTF) is significantly lower than Lyra v2, which processes audio approximately 10 times faster in real-time scenarios.
> Additionally, we evaluated **the most authoritative POLQA/P.863 objective metrics**, which are recommended by ITU to surpass PESQ.
>
> To make our assessment more complete and credible, we have supplemented the subjective evaluation results of DAC@8kbps, Encodec@12kbps, and SpeechTokenizer. We selected 10 subjects to conduct DCRMOS evaluation on 4 Mandarin corpora. Each subject compared the quality of the four systems and the reference audio, scoring them on a 1-5 scale. The results are shown in the following table. It should be emphasized that we have a streaming structure. Our subjective scores indicate that the competitive quality is achieved with the lowest delay and parameter quantity.
>
> Table Subjective evaluation on different neural speech codecs
> | **Codecs**  | VChangeCodec (Ours) |  DAC  | Encodec | SpeechTokenizer |
> | :---------: | :----------: | :---: | :-----: | :-------------: |
> | **Bitrate** |   9.5kbps    | 8kbps | 12kbps  |        -        |
> |  **MOS**  |     4.54     | 4.55  |  3.52   |      3.74       |
>
> For further clarification, you may refer to our detailed explanation of "Emphasizing our innovations in neural codec field and operator-oriented backgrounds" in the **general response**.
>
> > Q5: _**The synthetically generated parallel training data should be mentioned earlier.**_
>
> Thank you for your valuable suggestion. We have added this design rationale in our revised introduction.
> “The target timbre customization is achieved using near-parallel training data generated through open-source voice conversion toolkits."
>
> While our approach uses parallel training data, this design offers significant advantages through its simplicity and effectiveness. By decreasing the dependence on the speech recognition modules (as used in QuickVC [5], FreeVC [6] etc.), we substantially reduce model complexity while maintaining high performance. Moreover, our streamable architecture enables timbre adaptation for any speech input.
>
> [5] Guo H, Liu C, Ishi C T, et al. QUICKVC: A Lightweight VITS-Based Any-to-Many Voice Conversion Model using ISTFT for Faster Conversion[C]//2023 IEEE Automatic Speech Recognition and Understanding Workshop (ASRU). IEEE, 2023: 1-7.
>
> [6] Li J, Tu W, Xiao L. Freevc: Towards high-quality text-free one-shot voice conversion[C]//ICASSP 2023-2023 IEEE International Conference on Acoustics, Speech and Signal Processing (ICASSP). IEEE, 2023: 1-5.

---

> > ### Author Response · Authors · 2024-11-20
> > **Response to Reviewer UYJc (3/4)**
> >
> > > Q6: _**Evaluation of the voice change mode, provide subjective evaluation, compare these zero-shot conversion models.**_
> >
> > We appreciate the reviewer's suggestion about the importance of comprehensive evaluations.
> >
> > Importantly, we have provided additional clarification of "Regarding the differences with zero-shot VC solutions and the limits of target timbre" in **general response**. We hope that the reviewers will take time to peruse these insights. Additionally, we have added **a new target timbre 2 from the open-source VCTK dataset in a demonstration page at the following link**:
> >
> > https://anonymous666-speech.github.io/Demo-VChangeCodec/
> >
> > In this section, we provide a further explanation of our subjective experimental evaluations.
> > To enhance the credibility of our results and provide a more precise assessment of voice quality, we have supplemented the subjective evaluation of our voice change mode experiments by including **the latest VC benchmarks, DDDM-VC (AAAI 2024) and FACodec (ICML 2024)**. The evaluation involved ten subjects who assessed speech naturalness and speaker similarity on a 5-point mean opinion score (MOS) scale, ranging from 1 (bad) to 5 (excellent).
> >
> > We evaluated six VC systems, focusing on two target timbres (one male and one female).
> > The test set includes two male and three female speakers. This resulted in five conversion pairs for each specific target timbre, leading to a total of 30 converted utterances (five pairs * six systems) from the six VC systems evaluated by each subject.
> > Subjects scored naturalness (N-MOS) and similarity (S-MOS) for 30 converted utterances for the specific target timbre.
> > The subjective results are presented in the table below.
> >
> > Table N-MOS and S-MOS on Male timbre (Correspondence Table 2)
> >
> > |              | N-MOS | S-MOS | Resemblyzer (%) |
> > | :----------: | :---: | :---: | :-------------: |
> > |    VQMIVC    | 3.24  | 2.18  |      66.06      |
> > |   Diff-VC    | 2.94  | 2.60  |      79.74      |
> > |   QuickVC    | **4.05**  | 2.60  |     58.33      |
> > |   DDDM-VC    | 2.00  | 2.60  |      78.19      |
> > |   FACodec    | 2.73  | 2.82  |      81.08           |
> > | VChangeCodec (Ours) | 3.55  | **3.98**  |      **88.07**      |
> >
> > Table N-MOS and S-MOS on Female timbre (Correspondence Table 6)
> >
> > |              |  N-MOS   |  S-MOS   | Resemblyzer (%) |
> > | :----------: | :------: | :------: | :-------------: |
> > |    VQMIVC    |   2.88   |   2.17   |      52.70      |
> > |   Diff-VC    |   3.02   |   2.05   |      70.13      |
> > |   QuickVC    | **4.07** |   2.01   |      47.30      |
> > |   DDDM-VC    |   2.71   |   2.36   |      73.80      |
> > |   FACodec    |   3.35   |   2.61   |      74.11           |
> > | VChangeCodec (Ours)  |  4.00   | **4.38** |    **84.80**    |
> >
> > The experimental results demonstrate that our model achieves superior performance in S-MOS and attains near-optimal performance in N-MOS.
> > The subjective evaluation results not only align with our objective evaluation but demonstrate even stronger performance, particularly in speaker similarity (SMOS). This consistent performance across both objective and subjective measures reinforces the robustness of our approach.
> > It illustrates that our strategy is highly effective, delivering high-quality voice conversion for targeted applications.
> >
> > To enhance the credibility of our results, we conducted **objective evaluations by comparing our method with the latest DDDM-VC (AAAI 2024) and NaturalSpeech 3 (FACodec) system for specific target timbres**. This comparative analysis provides a comprehensive assessment of VC performance across different systems. The objective evaluation results are presented in the table below. Compared with the latest DDDM-VC and FACodec, our objective quality still remains competitive. Bold indicates the best result, while italicized values indicate the second-best results.
> >
> > Table Comparison with SOTA VC methods on Male timbre (Correspondence Table 2)
> >
> > |                     |   SIG    |   BAK    |   OVRL   |   MOS    |   MCD    |    WER     |    CER    | Resemblyzer |
> > | :-----------------: | :------: | :------: | :------: | :------: | :------: | :--------: | :-------: | :---------: |
> > |       VQMIVC        | **3.48** |   3.94   | **3.15** |   3.42   |   7.02   |   26.24%   |  14.56%   |   66.06%    |
> > |       Diff-VC       |   3.31   | **4.12** |   3.08   |  _3.70_  |   7.81   |   22.60%   |  11.15%   |   79.74%    |
> > |       QuickVC       |  _3.44_  |   3.89   |   3.07   |   3.68   |   7.01   | **10.22%** | **4.55%** |   58.33%    |
> > |       DDDM-VC       |   2.66   |   3.52   |   2.35   |   3.30   |   7.92   |   42.29%   |  25.02%   |   78.19%    |
> > |       FACodec       |   2.99   |   3.71   |   2.63   |   3.29   |  _6.87_  |   17.18%   |  10.33%   |  _81.08%_   |
> > | VChangeCodec (Ours) |   3.35   |  _4.11_  |  _3.11_  | **3.71** | **5.76** |  _16.19%_  |  _7.67%_  | **88.07%**  |
> > |  Oracles (Target)   |   3.29   |   4.04   |   3.06   |   3.84   |          |            |           |    100%     |

---

> > > ### Author Response · Authors · 2024-11-20
> > > **Response to Reviewer UYJc (4/4)**
> > >
> > > > Q6: _**Continue with the previous question Q6, unfair to compare these VC models.**_
> > >
> > > To comprehensively evaluate our model's competitiveness, we conducted fair experiments by **retraining selected baseline VC systems with our target timbre dataset** (Male timbre, correspondence Table 2). While Diff-VC and FACodec were excluded due to unavailable f0/spectrogram extraction scripts and training procedures respectively, we focused on three advanced systems: VQMIVC, QuickVC, and the recent DDDM-VC.
> > >
> > > Retraining details are as follows:
> > > VQMIVC: Trained from scratch for 500 epochs using extracted f0 and spectral features from target male timbre data.
> > > QuickVC: Fine-tuned for 3200k steps on their provided base model at 1200k steps  (due to significant training time requirements).
> > > DDDM-VC: Trained from scratch for 200k steps with offline-extracted f0.
> > >
> > > The objective evaluation results are presented in the table below.
> > > VQMIVC showed decreased performance, likely due to the need for retraining a dataset-specific vocoder, which needs more complexity.
> > > QuickVC demonstrated significant improvement in timbre similarity, though still not reaching our performance.
> > > DDDM-VC showed improvements across all objective metrics.
> > > Nevertheless, VChangeCodec maintains superior timbre adaptation capabilities and competitive voice quality. Crucially, our system achieves these results with **full streaming capability and a lightweight architecture of less than 1 million parameters**.
> > >
> > > We have substantially enhanced our paper with comprehensive evaluations and fair experimental settings. We would like to emphasize that the primary contribution of our work lies in the novel speech codec architecture.
> > > **We believe that this comprehensive experiment will alleviate your concerns.** If you have any further questions, please contact us.
> > >
> > >
> > > Table Comparison with the retrained/finetuned SOTA VC methods on Male timbre
> > >
> > > |                     |   SIG    |   BAK    |   OVRL   |   MOS    |   MCD    |    WER    |    CER    | Resemblyzer |
> > > | :-----------------: | :------: | :------: | :------: | :------: | :------: | :-------: | :-------: | :---------: |
> > > |       VQMIVC        | **3.46** |   3.82   |   3.03   |   2.95   |   6.58   |  118.96%  |  89.71%   |   56.61%    |
> > > |       QuickVC       |  _3.38_  | **4.11** | **3.16** | **3.74** |  _6.31_  | **9.07%** | **4.96%** |  _87.57%_   |
> > > |       DDDM-VC       |   2.21   |   3.28   |   1.96   |   3.34   |   6.73   |  29.49%   |  13.97%   |   83.00%    |
> > > | VChangeCodec (Ours) |   3.35   | **4.11** |  _3.11_  |  _3.71_  | **5.76** | _16.19%_  |  _7.67%_  | **88.07%**  |
> > > |  Oracles (Target)   |   3.29   |   4.04   |   3.06   |   3.84   |          |           |           |    100%     |
> > >
> > >
> > >
> > > > Q7: _**Resemblyzer is a weak alternative to perceptual evaluation.**_
> > >
> > > We have included the SMOS subjective evaluation in our experiments. Please refer to our response in Q6 for the details.
> > >
> > > > Q8: _**Notably, Casual projection network should be renamed into Causal projection network. When you say your model achieves superior latency it would mean it has a larger latency. you probably want to say your latency is lower than that of the other methods.**_
> > >
> > > We appreciate the reviewer's carefulness and suggestions. Upon double-checking, we have corrected any typos.
> > >
> > > We have refined the sentence "In summary, VChangeCodec demonstrates superior performance for VC, achieving lower latency, better MCD scores and higher speaker similarity compared to baseline methods. These comprehensive improvements make it particularly well-suited for real-time VC."
> > >
> > > > Q9: _**Please explain better how you perform the bit rate calculation in A.3 and specify the quantizer levels R that you use in your experiments.**_
> > >
> > > We have explained the bit rate calculation and the value of R. Please refer to our response in Q1 for the details.
> > >
> > > > Q10: _**All other VC models are operated in zero short mode. The results of your model are difficult to judge.**_
> > >
> > > We have substantially enhanced our paper with comprehensive evaluations and fair experimental settings in Q6.
> > > We believe that this comprehensive experiment will alleviate your concerns.
> > >
> > > Importantly, we have provided additional clarification of "Regarding the differences with zero-shot VC solutions and the limits of target timbre" in **general response**. We hope that the reviewers will take time to peruse these insights.
> > >
> > > > Q11: _**Why are neither DAC nor Encodec part of the perceptual evaluation in the original mode?**_
> > >
> > > We have responsed in Q4 for the details.
> > >
> > >
> > > > Q12: _**SimpleSpeech and SimpleSpeech 2. https://arxiv.org/pdf/2408.13893.**_
> > >
> > > Thanks for your reminder. In this paper, scalar quantization (SQ) is selected as the quantization method and we refer to the previous paper [7]. We will add extra references proposed by the reviewer in the revision.
> > >
> > > [7] Mentzer F, Minnen D, Agustsson E, et al. Finite scalar quantization: Vq-vae made simple[J]. arXiv preprint arXiv:2309.15505, 2023.

---

> ### Author Response · Authors · 2024-11-21
> **Request for Follow-Up on Rebuttal**
>
> Dear Reviewer UYJc:
>
> We have carefully responded to your queries and questions about our work. We have comprehensively enhanced the manuscript (new PDF revision) by providing **detailed descriptions of all parameters, datasets and neural codec models. We have added thorough subjective evaluations, including the latest VC baselines and our retrained model results to substantiate our conclusion**. Could you please let us know if our responses have addressed your concerns?
>
> If you feel we have successfully resolved your issues, we kindly request you consider **adjusting your initial score** accordingly. We hope to receive your reply, as your feedback is of significant value for the improvement of our work, and we would be immensely grateful.
>
> Please feel free to share any additional comments you may have.

---

> ### Author Response · Authors · 2024-11-23
>
> Dear Reviewer UYJc:
>
> We value active discussions with you. I hope you will take the time to read the rebuttal material, especially **comprehensive subjective evaluations, the latest VC baselines and our retrained model results to substantiate our conclusion**.
>
> If they resolve your concerns, we would greatly value an increase in the score.

---

> > ### Comment · Reviewer_UYJc · 2024-11-25
> >
> > Dear Authors
> >
> > I am sorry for the missing reaction to your numerous comments, I was busy with other things.
> >
> > I appreciate the many new evaluations you have added to your paper. I listened to the demo page, which appears to be very helpful because it allows everybody to put the different evaluations into perspective. Thanks for adding some missing details, notably concerning the bitrate calculation. I also very much appreciate the new evaluation using retrained versions of the other state-of-the-art methods. As expected the other state-of-the-art methods become much better. It is unfortunate that the new evaluations have all been put into the Annexe, and that all the discussions in the main part of the paper have been kept unchanged.
> >
> > I have to maintain my impression that the paper is not easy to understand and the new evaluations do not help here.
> > The mention (Correspondence table N) is unclear and requires the reader to jump through the document. I have the feeling that Table 8 (Correspondence 6) wrongly states in the caption that it is for the Male speaker, while if we go back through indirect references (Table 8 -> search Table 6 -> search text in Appendix 6) we find that  Table 6 is for the female speaker.
> > Anyway, given the innovative approach for the speaker conditioning, I will slightly change my review results.

---

> > > ### Author Response · Authors · 2024-11-25
> > > **Thank you for the reviewer's positive feedback**
> > >
> > > Dear Reviewer UYJc:
> > >
> > > Thank you so much for your thoughtful feedback! We're really happy that you've taken the time to review our rebuttal and updated manuscript, and we truly **value your new suggestions**.  We will try our best to reorganize our latest experimental results and rearrange the tables to make everything clearer and more accessible. **We'll submit the revised PDF as quickly as we can**.
> > >
> > > We're truly grateful for both your guidance and the updated score.

---

> ### Author Response · Authors · 2024-11-26
> **Please Review the Updated PDF Revision**
>
> Dear Reviewer UYJc:
>
> We appreciate your constructive suggestions about including new evaluations and all the discussions in the paper. **Your rigorous and detailed feedback has been helpful, it has truly motivated us to keep improving our manuscript.** We sincerely appreciate the time and effort you've put into reviewing our work. Considering the page limit, we have tried our best to reasonably incorporate new content into the main part of the paper. We sincerely invite you to take some time to check **the latest PDF revision**. We used **the blue font to highlight the new content**. We believe that **the new version can meet the requirements of the ICLR conference.**
>
> - First, we have added the discussion about motivation and differences with traditional VC in the introduction (in lines 83-87, 92-94 and 105-107).
>
> - To present our experimental results more comprehensively, we have integrated the original parameter table with Table 1. We also have added subjective evaluations and a discussion of speech codec (in lines 385-390).
>
> - We have added subjective evaluations of the voice change mode in Table 3 (in lines 432-447) and Table 8 (in lines 912-929). We also have provided a more detailed experimental discussion about objective evaluations (in lines 420-431).
>
> - Additionally, we have added the experiment of retrained VC models in Table 4 and the corresponding discussion (in lines 449-465).
>
> - We have also incorporated the suggestions of the other reviewers into the paper.
>
> We hope that all these revisions have improved **the clarity and presentation of our work**.
> These improvements significantly enhanced the manuscript's academic depth and comprehensiveness of our manuscript.
>
> We deeply value your detailed guidance on our work.
> **If the new PDF version successfully changes your impression, we sincerely appreciate your consideration for a higher score.** If you have any new questions, we would like to continue addressing them.
>
>
> Best regards,
>
> Authors.

---

> > ### Author Response · Authors · 2024-12-01
> > **Friendly reminder to review the latest response**
> >
> > Dear Reviewer UYJc:
> >
> > We sincerely thank you for your effort in reviewing our manuscripts. We kindly hope you can **find some time to review our latest responses and PDF revision**.
> >
> > As you requested, we have carefully integrated new subjective evaluations, all the discussions and clearer tables into the main body of the paper. We hope these substantial improvements will be considered in the final evaluation of our paper. We remain open to further discussion to enhance our work.
> >
> > Thank you once again for your constructive comments！
> >
> > Best regards,
> >
> > Authors.

---

### Official Review · Reviewer_Dy4A · 2024-11-04

**Soundness:** 2
**Presentation:** 3
**Contribution:** 3
**Rating:** 6
**Confidence:** 3

**Summary:**

This paper presents VChangeCodec, a codec specifically designed for real-time voice conversion. It uses a causal projection network to convert the vocal timbre of the source speaker into that of the target speaker at the token level. Evaluation results show that in original voice mode, VChangeCodec achieves good quality compared with other codec models. In voice change mode, it performs comparably to previous methods in terms of naturalness and intelligibility, and outperforms them in speaker similarity according to objective evaluations.

**Strengths:**

This paper presents a voice conversion system that can be used in real-time communication such as virtual meetings. The proposed module, the causal projection network, can be easily plugged into different codec frameworks and has very low latency. The evaluation results look reasonable based on the metrics used. The authors also conducted ablation studies to discuss the influence of each component in the proposed framework.

**Weaknesses:**

While there have been several works on zero-shot voice conversion (though they may not be real-time), the proposed framework is constrained to a predefined set of target speakers, which limits its extensibility. Additionally, the evaluation results have issues, such as using very limited test sets, inconsistency in the languages evaluated between the original and voice change modes, and a lack of human evaluation in the voice change mode.

**Questions:**

1. In Section 3.2, the authors imply that using pre-trained speaker embeddings would increase computational costs and storage space. However, if I understand correctly, the target speaker features are all pre-computed, so using speaker embeddings (which are usually single vectors) should not pose a problem. Moreover, speaker embeddings have been widely used in several speech synthesis tasks with good results. I wonder if the authors have conducted experiments using pre-trained speaker embeddings, and how their performance compares to the results presented in the paper.

2. In the evaluation, the authors used Mandarin utterances for assessing the original mode and English ones for the voice change mode. I am confused about this setting, as they are very different languages, and it is not natural to evaluate them in two different modes. It would make more sense to either (a) use only Mandarin or English, or (b) evaluate both the original and voice change modes in both English and Mandarin.

3. In the evaluation of the voice change mode, the authors only use automatic evaluations (if I understand correctly, the MOS scores in Tables 2 and 3 are not human ratings). It is necessary to include human evaluation results in speech synthesis tasks. The authors might first conduct objective evaluations on large test sets using automatic MOS models, and then perform human subjective evaluations on a randomly sub-sampled set to justify the reliability of the automatic MOS on the set.

4. Similar to point (3), the authors should conduct subjective evaluations on speaker similarity.

5. In Table 2, there are no values for oracles in the intelligibility column, which I believe the authors should include so that readers can understand the quality gap between the generated utterances and the authentic ones.

6. In Section 4.4, the authors compared the real-time factor of the proposed approach only with Lyra2. I believe there are other real-time voice conversion systems, such as StreamVC, and it would be good to include them.

7. I was unable to listen to the audio samples in the supplementary material. The authors included audio samples in a PPTX file, which is not directly accessible without installing Microsoft Office. I'm not sure if online converters break the audio links in the file, so I prefer not to use them. It would be helpful if the authors could provide the audio files directly.

---

> ### Author Response · Authors · 2024-11-19
> **Response to Reviewer Dy4A  (1/3)**
>
> Dear Reviewer Dy4A,
>
> Thank you so much for taking the time to review our paper and providing valuable comments and suggestions regarding the dataset description and human subjective evaluations. We will respond to your questions below.
>
> > W1: _**The proposed framework is constrained to a predefined set of target speakers, which limits its extensibility.**_
>
> We understand the reviewer’s concern about the extensibility of voice conversion (VC). We have provided additional clarification of "Regarding the differences with zero-shot VC solutions and the limits of target timbre" in **general response**. We hope that the reviewers will take the time to peruse these insights.
>
> We would like to further clarify our methodology and motivations (refer to **general response**) to highlight the key distinctions between our proposed approach and these zero-shot VC methods.
> Our main contribution lies in proposing an innovative method with **a solid embodiment and pipeline of the neural speech codec** with real-time VC features, rather than solely focusing on SOTA zero-shot VC methods.
> Our VChangeCodec is **a lightweight, low-latency codec tailored for mobile network operators**.
> We target deploying the codec through an operator (as detailed in lines 96-101).
> Given the practical constraints of latency and computational complexity in our target scenarios, our proposed method is expected to be integrated into mid-range performance smartphones.
>
> We need to clarify a few points regarding the design of a predefined set of target speakers:
>
> - First, we have carefully considered **privacy issues for applications in RTC services**.  Our voice conversion (VC) module is integrated into the speech communication system, prohibiting users from accessing or altering internal configurations.
> The embedded speaker representation is injected at the sending end and maintained by the operators, with designated timbres being pre-defined and inaccessible to ordinary users.
> In contrast, previous VC models, if positioned on the user side, would allow users to arbitrarily modify the pre-defined timbres before they pass through the sender's encoder. The converted timbre, then transmitted through the operator's codec, could raise issues of timbre infringement.
>
> - Secondly, the pre-trained weights of neural networks for specific target speakers are provided by the service provider (operator), and users must subscribe to the service from the operator. For example, after subscribing, a user might receive a binary file containing the weights for target speaker A, allowing them to change their voice to the timbre of speaker A but not to speaker B, etc. The number of target speakers in the service is controlled by the operator, who is also responsible for designing new target speakers to meet user requirements. Unlike SOTA zero-shot VC systems that **allow users to customize VC using just a few seconds of any target speaker's sample**, our approach with predefined speaker sets fundamentally minimizes privacy risks by avoiding the transmission and storage of arbitrary speaker characteristics.
>
> > Q1: _**The computational cost, storage space, and availability of experiments using pre-trained speaker embeddings.**_
>
> We appreciate the reviewer‘s deep understanding of the target speaker's features. Your understanding is correct regarding the vector of speech embeddings. However, the speaker extraction networks may be retrained on new corpora [1] or an adaptive network (average pooling) [2] is introduced, which will lead to additional training overhead.
> The openSMILE toolkit acquires a high performance against a large baseline feature set from the INTERSPEECH challenge [3]. The speaker embedding dimension is lower than the general extraction network 256, 512, etc. We have refined the description to enhance its rigor and clarity in our paper.
>
> Our paper primarily focuses on the codec architecture and its optimization. We found that OpenSmile's feature extraction adequately serves our current objectives and effectively contributes to timbre similarity while maintaining system simplicity. We acknowledge the valuable suggestion to explore alternative speaker embedding methods, we plan to conduct comprehensive comparisons with other speaker extraction methods (such as Wespeaker [4]) in future research.
>
> [1] Qian K, Zhang Y, Chang S, et al. Autovc: Zero-shot voice style transfer with only autoencoder loss[C]//International Conference on Machine Learning. PMLR, 2019: 5210-5219.
>
> [2] Yang Y, Kartynnik Y, Li Y, et al. StreamVC: Real-Time Low-Latency Voice Conversion[C]//ICASSP 2024-2024 IEEE International Conference on Acoustics, Speech and Signal Processing (ICASSP). IEEE, 2024: 11016-11020.
>
> [3] Eyben F, Scherer K R, Schuller B W, et al. The Geneva minimalistic acoustic parameter set (GeMAPS) for voice research and affective computing[J]. IEEE transactions on affective computing, 2015, 7(2): 190-202.
>
> [4] https://github.com/wenet-e2e/wespeaker

---

> > ### Author Response · Authors · 2024-11-19
> > **Response to Reviewer Dy4A (2/3)**
> >
> > > Q3: _**It is necessary to include human evaluation results in speech synthesis tasks. The authors might first conduct objective evaluations on large test sets using automatic MOS models, and then perform human subjective evaluations on a randomly sub-sampled set to justify the reliability of the automatic MOS on the set.**_
> >
> > We appreciate the reviewer's suggestion about the importance of human subjective evaluations.
> > We have provided additional experimental results in **general response**. In this section, we provide a further explanation of our subjective experimental evaluations.
> >
> > To enhance the credibility of our results and provide a more precise assessment of voice quality, we have supplemented the subjective evaluation of our voice change mode experiments by including the latest VC benchmarks, DDDM-VC (AAAI 2024) and FACodec (ICML 2024). The evaluation involved ten subjects who assessed speech naturalness and speaker similarity on a 5-point mean opinion score (MOS) scale, ranging from 1 (bad) to 5 (excellent).
> >
> > We evaluated six VC systems, focusing on two target timbres (one male and one female).
> > The test set includes two male and three female speakers. This resulted in five conversion pairs for each specific target timbre, leading to a total of 30 converted utterances (five pairs * six systems) from the six VC systems evaluated by each subject.
> > Subjects scored naturalness (N-MOS) and similarity (S-MOS) for 30 converted utterances for the specific target timbre.
> > The subjective results are presented in the table below.
> >
> > Table N-MOS and S-MOS on Male timbre (Correspondence Table 2)
> >
> > |              | N-MOS | S-MOS | Resemblyzer (%) |
> > | :----------: | :---: | :---: | :-------------: |
> > |    VQMIVC    | 3.24  | 2.18  |      66.06      |
> > |   Diff-VC    | 2.94  | 2.60  |      79.74      |
> > |   QuickVC    | **4.05**  | 2.60  |     58.33      |
> > |   DDDM-VC    | 2.00  | 2.60  |      78.19      |
> > |   FACodec    | 2.73  | 2.82  |      81.08           |
> > | VChangeCodec (Ours) | 3.55  | **3.98**  |      **88.07**      |
> >
> > Table N-MOS and S-MOS on Female timbre (Correspondence Table 6)
> >
> > |              |  N-MOS   |  S-MOS   | Resemblyzer (%) |
> > | :----------: | :------: | :------: | :-------------: |
> > |    VQMIVC    |   2.88   |   2.17   |      52.70      |
> > |   Diff-VC    |   3.02   |   2.05   |      70.13      |
> > |   QuickVC    | **4.07** |   2.01   |      47.30      |
> > |   DDDM-VC    |   2.71   |   2.36   |      73.80      |
> > |   FACodec    |   3.35   |   2.61   |      74.11           |
> > | VChangeCodec (Ours)  |  4.00   | **4.38** |    **84.80**    |
> >
> > The experimental results demonstrate that our model achieves superior performance in S-MOS and attains near-optimal performance in N-MOS.
> > The subjective evaluation results not only align with our objective evaluation but demonstrate even stronger performance, particularly in speaker similarity (SMOS). This consistent performance across both objective and subjective measures reinforces the robustness of our approach.
> > It illustrates that our strategy is highly effective, delivering high-quality voice conversion for targeted applications.
> >
> > Additionally, we would like to highlight that the DNSMOS (an objective evaluation) has gained broad acceptance in the field, as evidenced by its use in Soundstorm (arXiv 2024) [5] and StreamVC (ICASSP 2024) [6]. Similarly, Resemblyzer is utilized by both QuickVC and StreamVC.
> >
> > [5] Borsos Z, Sharifi M, Vincent D, et al. Soundstorm: Efficient parallel audio generation[J]. arXiv preprint arXiv:2305.09636, 2023.
> >
> > [6] Yang Y, Kartynnik Y, Li Y, et al. StreamVC: Real-Time Low-Latency Voice Conversion[C]//ICASSP 2024-2024 IEEE International Conference on Acoustics, Speech and Signal Processing (ICASSP). IEEE, 2024: 11016-11020.
> >
> > > Q4: _**Similar to point (3), the authors should conduct subjective evaluations on speaker similarity.**_
> >
> > We have incorporated the SMOS subjective evaluation into our analysis. Please refer to our response in Q3 for the details.

---

> > > ### Author Response · Authors · 2024-11-19
> > > **Response to Reviewer Dy4A (3/3)**
> > >
> > > > Q5: _**In Table 2, there are no values for oracles in the intelligibility column, which I believe the authors should include so that readers can understand the quality gap between the generated utterances and the authentic ones.**_
> > >
> > > Thank you for your suggestion regarding the oracle values in Table 2. We have added the objective MCD values, and for WER and CER, lower values indicate better performance, with a target of zero. The updated table is as follows.
> > >
> > >
> > > Table Comparison with SOTA VC methods on Male timbre (Correspondence Table 2)
> > >
> > > |                     |   SIG    |   BAK    |   OVRL   |   MOS    |   MCD    |    WER     |    CER    | Resemblyzer |
> > > | :-----------------: | :------: | :------: | :------: | :------: | :------: | :--------: | :-------: | :---------: |
> > > |       VQMIVC        | **3.48** |   3.94   | **3.15** |   3.42   |   7.02   |   26.24%   |  14.56%   |   66.06%    |
> > > |       Diff-VC       |   3.31   | **4.12** |   3.08   |  _3.70_  |   7.81   |   22.60%   |  11.15%   |   79.74%    |
> > > |       QuickVC       |  _3.44_  |   3.89   |   3.07   |   3.68   |   7.01   | **10.22%** | **4.55%** |   58.33%    |
> > > |       DDDM-VC       |   2.66   |   3.52   |   2.35   |   3.30   |   7.92   |   42.29%   |  25.02%   |   78.19%    |
> > > |       FACodec       |   2.99   |   3.71   |   2.63   |   3.29   |  _6.87_  |   17.18%   |  10.33%   |  _81.08%_   |
> > > | VChangeCodec (Ours) |   3.35   |  _4.11_  |  _3.11_  | **3.71** | **5.76** |  _16.19%_  |  _7.67%_  | **88.07%**  |
> > > |  Oracles (Target)   |   3.29   |   4.04   |   3.06   |   3.84   |   4.23   |     0.00%      |    0.00%      |    100%     |
> > >
> > >
> > > > Q6: _**In Section 4.4, the authors compared the real-time factor of the proposed approach only with Lyra2. I believe there are other real-time voice conversion systems, such as StreamVC, and it would be good to include them.**_
> > >
> > > We appreciate the contributions from StreamVC (published in ICASSP2024) regarding low-latency voice conversion (VC).
> > > While a direct RTF comparison is not feasible due to the unavailability of StreamVC's official implementation, we can conduct a fair comparison through algorithm delay metrics and model complexity. According to the original StreamVC paper, their system achieves a 60 ms algorithm delay and 70.8 ms end-to-end delay. In contrast, our VChangeCodec demonstrates superior efficiency with only 40 ms+ end-to-end delay.
> > > For model complexity, StreamVC builds upon the Soundstream architecture, which requires between 2.4 million and 8.4 million parameters. **Our approach is significantly more lightweight, utilizing less than 1 million parameters while achieving better latency performance**. These comparisons demonstrate that our method achieves lower latency and higher parameter efficiency.
> > >
> > > Furthermore, we would like to emphasize the distinctive novelty of VChangeCodec compared to StreamVC in our related work. While StreamVC requires modifying both ends of the communication protocol by transmitting modified tokens along with target speaker embeddings, our approach offers significant architectural advantages. VChangeCodec operates only at the encoding/sending side, elegantly combining semantic and timbre information into a single converted token. This design maintains full compatibility with existing decoders at the receiving end, eliminating the need for protocol modifications at the decoder side. Moreover, by avoiding the transmission of speaker-specific data, our approach not only ensures protocol compatibility but also fundamentally eliminates privacy risks in communication systems.
> > >
> > > >Q7: _**I was unable to listen to the audio samples in the supplementary material. The authors included audio samples in a PPTX file, which is not directly accessible without installing Microsoft Office. I'm not sure if online converters break the audio links in the file, so I prefer not to use them. It would be helpful if the authors could provide the audio files directly.**_
> > >
> > > Thank you for your valuable suggestion. In response, we have created a comprehensive webpage for demonstration and look forward to your listening. The page shows both original voice mode and voice change mode, including Mandarin and English utterances. Additionally, we have incorporated the latest VC system from 2024, DDDM-VC (AAAI, 2024) and NaturalSpeech 3 (FACodec, ICML 2024). If you have any questions, please feel free to contact us.
> > >
> > >  **We would greatly appreciate your time in comparing the subjective quality at the following link**:
> > >
> > > https://anonymous666-speech.github.io/Demo-VChangeCodec/

---

> ### Author Response · Authors · 2024-11-21
> **Request for Follow-Up on Rebuttal**
>
> Dear Reviewer Dy4A:
>
> We want to thank you once again for taking the time to review our manuscript!
>
> We value active discussions with you. I hope you will take the time to read the rebuttal material, especially **the clarification of the dataset, the comprehensive human subjective evaluation and the new demo link**. FYI, we have prepared **general response** to clarify our motivation and the limits of target timbre.
>
> If they resolve your concerns, **we would greatly value increaing the score**.

---

> > ### Comment · Reviewer_Dy4A · 2024-11-25
> >
> > Dear Authors,
> >
> > Thank you for your detailed responses, which address not only my concerns but also those of the other reviewers. I appreciate the effort you put into providing such thorough explanations. The audio demo sounds good to me, and I thank you for that as well.
> >
> > From your explanation, it seems that your scenario aligns more closely with an any-to-one voice conversion setup, where a specialized model is trained for each target speaker. If this is the case, I believe readers might naturally question why comparisons with such models were not included. Given the different scenarios, any-to-one models could be expected to perform better than any-to-any models, and they may also benefit from a potentially reduced number of parameters.
> >
> > I am willing to revise the Presentation and Contribution scores for now, and I look forward to your further clarification regarding my remaining concern.

---

> > > ### Author Response · Authors · 2024-11-25
> > > **Response to Reviewer Dy4A's question (1/2)**
> > >
> > > Dear Reviewer Dy4A:
> > >
> > > Thank you so much for your insightful feedback. It is highly appreciated that you took the time to review our rebuttal document, listen to our audio demo and gain a better understanding of our work. We appreciate the new points you've commented on, and we'd like to continue to address your questions.
> > >
> > > We acknowledge the reviewer's concerns, which are indeed valuable.
> > > You've understood correctly! Our method adheres to a "one model for one timbre" approach, utilizing a lightweight model tailored for deployment in mid-range performance smartphones. In general, our approach is a completely new approach targeting the real-time communication (RTC) scenario with distinctive technical requirements and design, in which the technical problems and requirements are quite different from generalized voice conversion methods.
> > >
> > > First, the key difference between our approach and traditional voice conversion (VC) models lies in their target audiences and operational contexts. **Our lightweight neural speech codec is embedded within RTC systems, where the encoder and decoder are immutable and maintained by mobile network operators**.
> > > Consequently, it is not reasonable to compare with any-to-one VC models on the aspect of the performance in VC only **but to compare the performance and the merit of our method, comprehensively**.
> > > In our opinion, the any-to-one VC cannot fulfill the RTC requirements. The transmission process of any-to-one VC in RTC is shown in Figure 1 (a), placing a lightweight VC model either as a pre-processor or post-processor of the codec (e.g., VC-->Codec or Codec-->VC) would inevitably introduce double latency for both VC processing and compression.
> > > For example, the lightest AC-VC model [1] exhibits an algorithmic **delay of 57.5 ms and at least two million parameters** when running on a CPU infrastructure. Considering the additional 10 ms delay of the LPCNet pitch predictor in this system, and the 40 ms delay of the speech codec, the delay will reach **a cumulative latency of over 107.5 ms**.
> > > This significantly exceeds the acceptable latency threshold for RTC requirements.
> > >
> > > Secondly, our approach fundamentally differs from this lightweight VC model in methodology and efficiency. Instead of treating VC as a separate process, we integrate timbre adaptation directly within the codec transmission pipeline in Figure 1 (b). By incorporating a causal projection network into the existing speech codec architecture, our method maintains **just a 40 ms algorithmic delay without introducing any additional latency and less than one million parameters**.
> > > This novel framework represents a paradigm shift from traditional cascaded VC-codec systems to an integrated, efficient solution that achieves timbre adaptation and compression in a single unified pipeline.
> > >
> > > Thirdly, we have carefully considered privacy issues for applications in RTC services. Our VC module is integrated into the speech communication system, prohibiting users from accessing or altering internal configurations. The embedded speaker representation is injected at the sending end and maintained by the operators, with designated timbres being pre-defined and inaccessible to ordinary users. A potential use case is expected the user can only download the related binary file after subscribing to a specific target timbre from the operator. Consequently, the user cannot arbitrarily change the source voice to any target timbre to minimize the deepfake risk to operators.
> > >
> > >
> > > [1] Ronssin D, Cernak M. AC-VC: non-parallel low latency phonetic posteriorgrams based voice conversion[C]//2021 IEEE Automatic Speech Recognition and Understanding Workshop (ASRU). IEEE, 2021: 710-716.

---

> > > > ### Author Response · Authors · 2024-11-25
> > > > **Response to Reviewer Dy4A's question (2/2)**
> > > >
> > > > To comprehensively evaluate our model's competitiveness, we have experimented by **retraining selected baseline VC systems with our target timbre** (Male timbre, correspondence Table 2). **Such experiments can be viewed as an equivalent comparison to any-to-one models**. While Diff-VC and FACodec were excluded due to unavailable f0/spectrogram extraction scripts and training procedures respectively, we focused on three advanced systems: VQMIVC, QuickVC, and the recent DDDM-VC.
> > > >
> > > > Retraining details are as follows:
> > > > VQMIVC: Trained from scratch for 500 epochs using extracted f0 and spectral features from target male timbre data.
> > > > QuickVC: Fine-tuned for 3200k steps on their provided base model at 1200k steps  (due to significant training time requirements).
> > > > DDDM-VC: Trained from scratch for 200k steps with offline-extracted f0.
> > > >
> > > > The objective evaluation results are presented in the table below. Bold indicates the best result, and italics indicate the second-best result.
> > > > VQMIVC showed decreased performance, likely due to the need for retraining a dataset-specific vocoder, which needs more complexity. Considering the limited time, we did not retrain the vocoder.
> > > > QuickVC demonstrated significant improvement in timbre similarity, though still not reaching our performance.
> > > > DDDM-VC showed improvements across all objective metrics.
> > > > Nevertheless, VChangeCodec maintains superior timbre adaptation capabilities and competitive voice quality. Crucially, our system achieves these results with **full streaming capability and a lightweight architecture of less than 1 million parameters**.
> > > >
> > > > We have substantially enhanced our paper with comprehensive evaluations. We would like to emphasize that the primary contribution of our work lies in the novel speech codec architecture.
> > > > **We believe that this comprehensive experiment will alleviate your concerns.**
> > > >
> > > >
> > > > Table Comparison with the retrained/finetuned SOTA VC methods on Male timbre
> > > >
> > > > |                     |   SIG    |   BAK    |   OVRL   |   MOS    |   MCD    |    WER    |    CER    | Resemblyzer |
> > > > | :-----------------: | :------: | :------: | :------: | :------: | :------: | :-------: | :-------: | :---------: |
> > > > |       VQMIVC        | **3.46** |   3.82   |   3.03   |   2.95   |   6.58   |  118.96%  |  89.71%   |   56.61%    |
> > > > |       QuickVC       |  _3.38_  | **4.11** | **3.16** | **3.74** |  _6.31_  | **9.07%** | **4.96%** |  _87.57%_   |
> > > > |       DDDM-VC       |   2.21   |   3.28   |   1.96   |   3.34   |   6.73   |  29.49%   |  13.97%   |   83.00%    |
> > > > | VChangeCodec (Ours) |   3.35   | **4.11** |  _3.11_  |  _3.71_  | **5.76** | _16.19%_  |  _7.67%_  | **88.07%**  |
> > > > |  Oracles (Target)   |   3.29   |   4.04   |   3.06   |   3.84   |     4.23     |     0.00%      |     0.00%      |    100%     |
> > > >
> > > >
> > > > Thank you for raising the scores for Presentation and Contribution. We're committed to addressing all your new concerns thoroughly. **If we can successfully clarify these points, we hope you might consider further improving the overall score**. Please don't hesitate to ask if anything needs further clarification.

---

> > > > > ### Comment · Reviewer_Dy4A · 2024-11-25
> > > > >
> > > > > Dear Authors,
> > > > >
> > > > > Thank you for your response. I believe these explanations should be included in the paper, though I understand space constraints might make that challenging. I will slightly adjust my review score.

---

> > > > > > ### Author Response · Authors · 2024-11-25
> > > > > > **Response and appreciation to reviewer's feedback**
> > > > > >
> > > > > > Dear Reviewer Dy4A:
> > > > > >
> > > > > > Thank you for your prompt response and the updated scores. **We're truly encouraged by your feedback!** We are **working hard to incorporate these new explanations into our paper** and will submit the revised PDF version as soon as possible.
> > > > > >
> > > > > > Your suggestions have been incredibly helpful, and we welcome any additional feedback you might have. We sincerely appreciate your valuable questions about our work.
> > > > > >
> > > > > > Best regards,
> > > > > >
> > > > > > Authors.

---

> ### Author Response · Authors · 2024-11-23
>
> Dear Reviewer Dy4A:
>
> We hope we've successfully addressed all your concerns, and in this case, we would be happy if the reviewer could kindly consider increasing the score. If not, we're pleased to continue our dialogue and provide any additional clarifications you feel would be helpful.

---

> ### Author Response · Authors · 2024-11-27
> **Invite reviewer Dy4A to read the updated PDF revision**
>
> Dear Reviewer Dy4A:
>
> Thank you again for your valuable suggestions and guidance throughout this review process. Given the page constraints, we have carefully integrated all essential explanations into the main body of the paper. **We sincerely invite you to take another look at our revised PDF**. You should be able to see it by clicking the top-right PDF button. We used **the blue font to highlight the new content**. Thanks to your helpful feedback, our manuscript has been substantially enhanced. Specifically, we have made the following key improvements:
>
> - We have added explanations about the difference from generalized VC in the introduction (in lines 83-87 and 92-94) and have included a demonstration link for better accessibility (in lines 112-113).
>
> - We have supplemented our target scenarios and privacy issues of applications in RTC services (in lines 772-796).
>
> - We have added the comprehensive dataset descriptions and experimental setup details (in lines 301-304 and 864-876).
>
> - We have added subjective experiments in Table 3 (in lines 432-447) and Table 8 (in lines 912-929). We also have provided a more detailed discussion of objective evaluation results (in lines 420-431).
>
> - Additionally, we have added the experiment on retrained VC models (any-to-one) in Table 4 and the corresponding analysis (in lines 449-465).
>
> - We have also integrated valuable suggestions from other reviewers.
>
> These changes not only make our work more thorough and convincing but also substantially strengthen the academic rigor of our manuscript. We believe these improvements have effectively enhanced the overall quality of our paper.
>
> **We hope we have resolved all the concerns and that the revised manuscript will better suit the ICLR conference.** If so, we would be very happy if you could kindly consider increasing the score. If not, we are very willing to engage in a discussion with you. We will do our best to answer any other questions and give more explanations.
>
> Best regards,
>
> Authors.

---

### Author Response · Authors · 2024-11-17
**General Response to all reviewers (1/3)**

Dear reviewers,

We sincerely thank the reviewers for their valuable insights and constructive suggestions.

It is encouraging to us that **all reviewers** highlight the low-latency goal of our architecture, recognizing it as a strong and promising solution for real-time communication (RTC).
Moreover, several reviewers agree that our VChangeCodec is innovative and interesting, with a highly ambitious target (Y2Se, UYJc). The proposed causal projection network can be seamlessly incorporated into different codec frameworks (Dy4A, UYJc).
Additionally, our paper is well-written and well-organized, which contributes to its readability (Y2Se, 2ct9).

We have carefully addressed each of the key concerns raised by the reviewers to enhance the understanding and clarification of our innovative contributions to the **speech coding community and real-time communications (RTC)**.
We have provided additional experimental results, including comparisons with state-of-the-art (SOTA) neural speech codecs and voice conversion (VC) methods, along with a new demonstration page. **We would greatly appreciate your time in comparing the subjective quality**.

> Emphasizing our innovations in neural codec field and operator-oriented backgrounds (for Y2Se, Dy4A, 2ct9).

Our VChangeCodec is **a lightweight, low-latency codec tailored for mobile network operators**.
We target deploying the codec through an operator rather than via peer-to-peer communication (as detailed in lines 96-101).
Given the practical constraints of latency and computational complexity in our target scenarios, our proposed method is expected to be integrated into mid-range performance smartphones.


- First, the RTC **speech coding standards** widely applied in telecommunication and internet society are Opus and EVS (traditional signal processing-based) with high quality of experience including high quality, low latency and low complexity.
To substantiate that our proposed method is competitive, we have compared it with these established standards to demonstrate that it offers comparable quality, latency, and complexity. A gentle reminder that other neural coding methods, such as Soundstream and Encodec, have similarly been benchmarked against these established standards in their experiments.

- Secondly, our method has strong RTC features (employing a fully streaming architecture) to fulfill the requirements in low complexity, which is not achievable with codecs like Descript Audio Codec (DAC, Neurips 2024), FACodec (ICML 2024), and SpeechTokenizer (ICLR 2024) in their current designs.
For example, the DAC codec has 75 million parameters, and the FACodec has 1 billion parameters. According to the latest codec evaluation [1], the performance gains obtained by DAC et al. may come with increased complexity.

- Furthermore, using a lightweight voice conversion (VC) model as either a post-processor or pre-processing module in the codec chain (i.e., VC-->Codec or Codec-->VC) would inherently double the latency for both VC processing and compression.
For example, the lightest AC-VC model to our knowledge [2], when run on a CPU, exhibits an algorithmic delay of 57.5 ms, along with the 40 ms delay from the speech codec, the cumulative delay exceeds 97.5 ms, which is deemed unacceptable for RTC requirements.

- In fact, our proposed method can provide prominent performance in both original voice compression quality and voice conversion quality with the parameter amount of **less than 1 million and the low delay of just 40 ms**.
Our main contribution lies in proposing an innovative method with a solid embodiment and pipeline of **the speech codec with additional/optional VC features**, rather than solely focusing on SOTA zero-shot VC methods. The merit of our approach is particularly distinctive for the target scenario defined.

[1] Muller T, Ragot S, Gros L, et al. Speech quality evaluation of neural audio codecs[C]//Proc. Interspeech. 2024.

[2] Ronssin D, Cernak M. AC-VC: non-parallel low latency phonetic posteriorgrams based voice conversion[C]//2021 IEEE Automatic Speech Recognition and Understanding Workshop (ASRU). IEEE, 2021: 710-716.

---

> ### Author Response · Authors · 2024-11-17
> **General Response to all reviewers (2/3)**
>
> > Regarding to provide the complete subjective evaluations (for all reviewers).
>
> Regarding the subjective evaluation of Codec, we have supplemented the subjective evaluation results of DAC@8kbps, Encodec@12kbps, and SpeechTokenizer. We invited ten subjects to conduct a DCRMOS evaluation on four Mandarin corpora. Each subject compared the quality of  speech compressed by four systems with the reference speech, scoring them on a 1-5 scale.
>
> The results are shown in the following table. The test result indicated clearly the competitive quality is achieved in the method we proposed even it has a streaming structure with the lowest delay and parameter quantity.
>
> Table Subjective evaluation on different neural speech codecs
> | **Codecs**  | VChangeCodec (Ours) |  DAC  | Encodec | SpeechTokenizer |
> | :---------: | :----------: | :---: | :-----: | :-------------: |
> | **Bitrate** |   9.5kbps    | 8kbps | 12kbps  |        -        |
> |  **MOS**  |     4.54     | 4.55  |  3.52   |      3.74       |
>
>
> To more accurately represent voice quality and make our results more convincing, we have supplemented the subjective evaluation of our voice change mode experiments by including the latest VC benchmarks, DDDM-VC (AAAI 2024) and FACodec (ICML 2024). The evaluation involved ten subjects who assessed speech naturalness and speaker similarity on a 5-point mean opinion score (MOS) scale, ranging from 1 (bad) to 5 (excellent).
>
> We evaluated six VC systems, focusing on two target timbres (one male and one female).
> The test set includes two male and three female speakers. This resulted in five conversion pairs for each specific target timbre, leading to a total of 30 converted utterances from the six VC systems evaluated by each subject. Subjects scored naturalness (NMOS) and similarity (SMOS) for 30 converted utterances for the specific target timbre. The subjective results are presented in the table below.
>
> Table N-MOS and S-MOS on Male timbre (Correspondence Table 2)
>
> |              | N-MOS | S-MOS | Resemblyzer (%) |
> | :----------: | :---: | :---: | :-------------: |
> |    VQMIVC    | 3.24  | 2.18  |      66.06      |
> |   Diff-VC    | 2.94  | 2.60  |      79.74      |
> |   QuickVC    | **4.05**  | 2.60  |     58.33      |
> |   DDDM-VC    | 2.00  | 2.60  |      78.19      |
> |   FACodec    | 2.73  | 2.82  |      81.08           |
> | VChangeCodec (Ours) | 3.55  | **3.98**  |      **88.07**      |
>
> Table N-MOS and S-MOS on Female timbre (Correspondence Table 6)
>
> |              |  N-MOS   |  S-MOS   | Resemblyzer (%) |
> | :----------: | :------: | :------: | :-------------: |
> |    VQMIVC    |   2.88   |   2.17   |      52.70      |
> |   Diff-VC    |   3.02   |   2.05   |      70.13      |
> |   QuickVC    | **4.07** |   2.01   |      47.30      |
> |   DDDM-VC    |   2.71   |   2.36   |      73.80      |
> |   FACodec    |   3.35   |   2.61   |      74.11           |
> | VChangeCodec (Ours)  |  4.00   | **4.38** |    **84.80**    |
>
> Based on our findings, we can confidently state that our model attains the highest scores in subjective evaluation for S-MOS and the near-optimal performance in N-MOS. Our method is particularly tailored to specific target timbres, offering less versatility in timbre conversion but superior quality in the conversion of selected timbres. This approach illustrates that the "one timbre, one model" strategy is not only feasible but also robust.
>
> Additionally, we would like to highlight that the DNSMOS (an objective evaluation) has gained broad acceptance in the field, as evidenced by its use in Soundstorm (arXiv 2024) [3] and StreamVC (ICASSP 2024) [4]. Similarly, Resemblyzer is utilized by both QuickVC and StreamVC. Furthermore, the Whisper recognition model stands as the official method for assessing intelligibility in the Codec-SUPERB competition [5] and is also adopted by SpeechTokenizer (ICLR 2024).
>
> [3] Borsos Z, Sharifi M, Vincent D, et al. Soundstorm: Efficient parallel audio generation[J]. arXiv preprint arXiv:2305.09636, 2023.
>
> [4] Yang Y, Kartynnik Y, Li Y, et al. StreamVC: Real-Time Low-Latency Voice Conversion[C]//ICASSP 2024-2024 IEEE International Conference on Acoustics, Speech and Signal Processing (ICASSP). IEEE, 2024: 11016-11020.
>
> [5] https://github.com/voidful/Codec-SUPERB
>
>
>  > Regarding the new demo link for our VChangeCodec (for all reviewers):
>
> To facilitate a more direct assessment of audio quality, we have updated an anonymous GitHub repository with a demo. Notably, we have incorporated two new voice conversion (VC) systems, DDDM-VC and NaturalSpeech 3 (FACodec) from 2024, along with English and Mandarin corpora for our VChangeCodec. **We invite the reviewers to listen to the samples available at the following link**:
>
> https://anonymous666-speech.github.io/Demo-VChangeCodec/

---

> ### Author Response · Authors · 2024-11-17
> **General Response to all reviewers (3/3)**
>
> > Regarding the differences with zero-shot VC solutions and the limits of target timbre (for 2ct9, Dy4A).
>
> - We have carefully considered privacy issues for applications in RTC services. Unlike zero-shot VC, which allows users to select any target speaker's timbre in online conversations, our method restricts this flexibility. It's important to note that our voice conversion (VC) module is integrated into the speech communication system, prohibiting users from accessing or altering internal configurations. Consequently, users cannot freely change voices within an RTC system. In practical applications, the service operator should maintain model parameters on the backend, and users (on the sending side) must download a specific model corresponding to a certain timbre after subscribing.
>
> - Zero-shot VC operates on a "one model for all timbres" principle and is not optimized for low latency or low complexity on mobile devices. In contrast, our method adheres to a "one model for one timbre" approach, utilizing a lightweight model tailored for deployment in mid-range performance smartphones. Therefore, our approach is a completely new approach targeting the RTC scenario with distinctive technical requirements and design.
>
> - Conclusively, the proposal in this paper aims to enhance the experience in real-time communication, including conversational chatting. Therefore, the extensibility of the target speaker should be properly considered due to privacy concerns. This consideration is a cornerstone of our contribution, and to the best of our knowledge, this represents the first disclosure of a paper addressing this specific aspect.
>
> > Regarding the dataset description in detail (for Dy4A, UYJc).
>
> - We provide the following clarifications on the evaluation corpus.
> For the objective evaluation of the original voice mode, we utilized both Mandarin and English utterances.
> For the subjective evaluation (as detailed in lines 347-351), we focused on Mandarin utterances, given the subjects’ native language.
> We have provided both English and Mandarin utterances of the original mode in our anonymous repository.
>
> - In the voice change mode evaluation, we assessed both Mandarin and English utterances. Notably, we chose to evaluate English utterances using Word Error Rate (WER) and Character Error Rate (CER) exclusively, as the Whisper model is less applicable to Mandarin, thus ensuring a more equitable comparison.
> For DNSMOS, MCD, and Resemblyzer assessments, we included both Mandarin and English utterances. We have bolstered our evaluation with the addition of N-MOS and S-MOS to substantiate our results.
>
> - Regarding the training set in original voice mode, we used all DNS challenge 2020 dataset.
> Recognizing that neural speech coding is inherently data-driven, we incorporated mixed speech segments, including those with noise or music, to enhance the robustness of our scheme. Our test set is strictly an out-of-domain dataset, ensuring it was not exposed to any model during training. In line with the ITU-T P.800 recommendation and adhering to the codec evaluation standard [1], our test set is used to assess all codec models on an identical, unseen test corpus.
>
> [1] Muller T, Ragot S, Gros L, et al. Speech quality evaluation of neural audio codecs[C]//Proc. Interspeech. 2024
>
> Please let us know if you have any further questions. Thank you!
>
> **Demo link**
>
> Dear ACs,
>
> Following the rebuttal guideline email, we provided an anonymous link to our demo sample to show the results in our paper:
> https://anonymous666-speech.github.io/Demo-VChangeCodec/.
>
> We will upload a **new PDF file** (revision) according to the reviewers’ valuable suggestions. We appreciate your ongoing interest in our research. In the following, under each reviewer's comment, **we address the concerns of the reviewers point by point**.
>
> Best regards,
>
> The 10137 authors.

---

### Author Response · Authors · 2024-11-21
**Author Response & Revision & Active Discussions**

Dear Reviewers,

We appreciate your helpful, insightful, and constructive comments on our paper.
We have carefully and comprehensively responded to each reviewer's feedback point by point.
**We value active discussions with all reviewers. I hope you will take the time to review the rebuttal material and provide feedback.**

We have supplemented various experiments to address the reviewers' concerns. Please check the **updated pdf file**.
For clarity, the new sections are currently located in the Appendix.
We hope the revised manuscript will better meet ICLR's academic standards. We are happy to discuss with the reviewers and consider further revisions.

For all reviewers's suggestions, we have included the subjective evaluations of the target timbre in Tables 7 and Table 8. (Appendix A.7)

Following Reviewer Dy4A's suggestions, we added a new demo page and dataset clarifications and descriptions in section 4.1.

Following Reviewer UYJc's suggestions, we added a detailed dataset description,  neural speech codec baseline information, subjective evaluations of DAC and Encodec, and the results of retrained VC models. (Appendix A.5 and A.9 - Experimental details)

Following Reviewer Y2Se's suggestion, we added references. (Appendix A.10)

Following Reviewer 2ct9's suggestions, we added new baselines  DDDM-VC (AAAI, 2024) and NaturalSpeech 3 (FACodec, ICML 2024) in Table 2 and Table 6.

Therefore, these improvements significantly enrich the manuscript's academic depth and research comprehensiveness.

---

> ### Author Response · Authors · 2024-11-28
> **Please Review the Updated PDF**
>
> Dear Reviewers,
>
> We have thoroughly incorporated all valuable suggestions into the main body of the paper,  including our motivation, comprehensive experimental results and detailed discussions. We sincerely invite you to take some time to read our revised PDF. You should be able to see it by clicking the top-right PDF button. We used the blue font to highlight the new content.
>
> We would greatly appreciate it if you could review these updates. Your feedback would be invaluable for further improving our work.
>
> Best regards,
>
> Authors

---

### Author Response · Authors · 2024-11-24
**Hope the reviewers will take note of our response**

Dear Reviewers,

Since receiving your initial feedback, we've worked hard to address your suggestions by revising our paper and conducting the additional experiments you recommended. We'd value **the chance to interact with you during the discussion**.

We would like to thank the reviewers again for their time and effort in reviewing our manuscript.

Kind regards,

Authors

---

### Meta-Review · Area_Chair_hycw · 2024-12-19

**Metareview:**

This paper presents an approach that combines a voice codec with an integrated identity conversion component. It achieves real-time voice identity conversion with low latency, making it suitable for applications like virtual meetings. The paper is well-organized, with a meticulously written background and methods section that clearly explains the concepts. The authors have evaluated their models using both objective and subjective metrics, demonstrating either improved performance or matching state-of-the-art codecs and voice conversion models.

However, the proposed framework is constrained to a predefined set of target speakers, limiting its extensibility compared to other zero-shot voice conversion works. The motivation for integrating the voice conversion model directly into the codec framework is unclear, especially when a lightweight voice conversion model could be used as a post-processor or pre-processor. Apart from the combination of the Codec+VC module and the scalar quantization trick, the paper lacks significant novelty. The paper should present the differences between VChangeCodec and related works, and include a comparison in the experimental evaluation.

**Additional Comments On Reviewer Discussion:**

During the author-reviewer discussion, the paper underwent a thorough review, which increased the overall rating. However, it still remains below the threshold of ICLR publications.

---

### Decision · Program_Chairs · 2025-01-22

Reject